

# Constraining the budget of NOx and VOCs at a remote Tropical island using multi-platform observations and WRF-Chem model simulations

Catalina Poraicu[1], Jean-François Müller[1], Trissevgeni Stavrakou[1], Crist Amelynck[1,2], Bert W. D. Verreyken[1,2,3,4], Niels Schoon[1], Corinne Vigouroux[1], Nicolas Kumps[1], Jérôme Brioude[3], Pierre Tulet[3,5], Camille Mouchel-Vallon[5,6]

1 Royal Belgian Institute for Space Aeronomy (BIRA-IASB), Ringlaan 3, 1180 Brussels, Belgium
2 Department of Chemistry, Ghent University, 9000 Ghent, Belgium
3 Laboratoire de l'Atmosphère et des Cyclones (LACy), UMR 8105, CNRS, Université de La Réunion, Météo France, 97744 Saint-Denis, France
4 University of Liège - Gembloux Agro-Biotech, Biosystems Dynamics and Exchanges (BIODYNE), 8 Avenue de la faculté, 5030 Gembloux, Belgium
5 Laboratoire d'Aérologie (LAERO), UMR 5560, CNRS, Université Paul Sabatier, IRD, Toulouse, 31400, France
6 Barcelona Supercomputing Center (BSC), Barcelona, Spain

*Correspondence to*: Catalina Poraicu (catalina.poraicu@aeronomie.be)

**Abstract.** Volatile organic compounds (VOCs) act as precursors to ozone and secondary organic aerosols, which have significant health and environmental impacts, and they can reduce the atmospheric oxidative capacity. However, their budget remains poorly quantified, especially over remote areas such as the Tropical oceans. Here, we present high-resolution simulations of atmospheric composition over Réunion Island, in the Indian Ocean, using the Weather Research and Forecasting model coupled with Chemistry (WRF-Chem). The coexistence and spatial heterogeneity of anthropogenic and biogenic emission sources in this region present a valuable but challenging test of the model performance. The WRF-Chem model is evaluated against several observational datasets, including Proton Transfer Reaction Mass Spectrometry (PTR-MS) measurements of VOCs and oxygenated VOCs (OVOCs) at the Maïdo Observatory (2160m a.s.l.) in January and July 2019. While the primary goal of our study is a better understanding of (O)VOC budget at remote Tropical latitudes, important model refinements are made to improve the model performance, including the implementation of high-resolution anthropogenic and biogenic isoprene emissions, updates to the chemical mechanism, and adjustments to the boundary conditions. These refinements are supported by comparisons with PTR-MS data as well as with meteorological measurements at Maïdo, in situ NOx and $O_3$ measurements from the air quality Atmo-Réunion network, Fourier Transform Infrared Spectroscopy (FTIR) measurements of $O_3$, CO, ethane and several OVOCs, also at Maïdo, and satellite retrievals from the TROPOspheric Monitoring Instrument (TROPOMI).
TROPOMI $NO_2$ data suggests that anthropogenic emissions, particularly from power plants near Le Port, dominate NOx levels over the island. Both TROPOMI and in situ surface $NO_2$ comparisons are used to adjust the power plant emissions at Le Port. Surface ozone concentrations are overestimated by ~6 ppbv on average, likely due to the neglect of halogen



chemistry in the model. Whereas modelled $NO_2$ over oceans is too low in summer when the lightning source is turned off, the inclusion of this source results in model overestimations corroborated by comparisons with upper tropospheric $NO_2$ mixing ratios derived from TROPOMI using the cloud-slicing technique (Marais et al., 2021). The model generally succeeds in reproducing the PTR-MS isoprene and its oxidation products (Iox), except for a moderate underestimation (~30%) of noontime isoprene concentration and for modelled concentration peaks near dawn and dusk, that are not seen in the observations. The ratio of Iox to isoprene (0.8 at noon in January) is fairly well reproduced by the model. The methanol and monoterpenes observations both suggest overestimations of their biogenic emissions, by factors of about 2 and 5, respectively. Acetaldehyde anthropogenic emissions are likely strongly overestimated, due to the lumping of higher aldehydes into this compound. Without this lumping, the modelled acetaldehyde would be underestimated by almost one order of magnitude, suggesting the existence of a large missing source, likely photochemical. The comparisons suggest the existence of a biogenic source of MEK equivalent to about 3% of isoprene emissions, likely associated with the dry deposition and conversion to MEK of key isoprene oxidation products. A strong model underestimation of the PTR-MS signal at mass 61 is also found, by a factor of 3-5 during daytime, consistent with previously reported missing sources of acetic and peracetic acid.

# 1 Introduction

Volatile organic compounds (VOCs) play a key role in important chemical processes throughout the atmosphere. They are precursors of ozone and secondary organic aerosols, both responsible for negative effects on human health (Jerrett et al., 2009; Pye et al., 2021) and the environment (e.g. Sicard et al., 2017), while also having climatic consequences (Mickley et al., 2004; Kanakidou et al., 2005; Shindell et al., 2006; Shrivastava et al., 2017). VOCs are mostly emitted by biogenic, anthropogenic and pyrogenic sources. The emissions of biogenic VOCs (BVOCs) due to terrestrial vegetation are affected by meteorological conditions (Guenther et al., 2006). VOCs undergo oxidation mainly by reaction with the hydroxyl radical (OH). OH is responsible for the removal of numerous pollutants in the atmosphere, initiating oxidation processes that usually transform airborne species into more oxygenated, and therefore more water soluble compounds (Comes, 1994). The reaction between VOCs and OH can effectively deplete OH and diminish the oxidative capacity of the atmosphere, thereby increasing the lifetime of pollutants and greenhouse gases such as methane (Zhao et al., 2019). Following the initial reaction of a VOC with OH, the further degradation of oxidation products can lead to additional OH loss, especially in remote regions with low levels of NOx (NOx = NO + $NO_2$) (Di Carlo et al., 2004; Read et al., 2012; Travis et al., 2020), where the reaction with NO is not the dominant sink of peroxy radicals (Logan, 1985; Atkinson, 2000).

Field measurements of OH reactivity in the remote troposphere have revealed the existence of a "missing" OH sink, primarily attributed to unknown organic compounds, in particular over boreal (Sinha et al., 2010) and tropical forests (Nölscher et al., 2016; Pfannerstill et al., 2021), as well as over the remote marine boundary layer (MBL) (Thames et al., 2020). In addition, models underestimate the concentrations of several known oxygenated VOCs (OVOCs) contributing to



OH reactivity over the remote ocean, most importantly acetaldehyde (Travis et al., 2020). The high observed abundances of acetaldehyde were recently supported by unexpectedly high measured concentrations of peroxyacetic acid (PAA) over the remote ocean (Wang et al., 2019), since PAA is photochemically produced almost exclusively from acetaldehyde oxidation under low NOx conditions.  The underestimation of acetaldehyde in models (e.g. Millet et al., 2010; Read et al., 2012; Travis et al., 2020) indicates a missing source likely due to air/sea exchange and secondary photochemical formation from oceanic precursors (Singh et al., 2003; Millet et al., 2010; Wang et al., 2019). Acetone is a substantial source of HOx radicals (HOx=OH+HO$_2$) in the upper troposphere and lower stratosphere (UT/LS) (Müller and Brasseur, 1999; Wang et al., 2020). Like acetaldehyde, it is a compound strongly regulated by the ocean, but which appears to be better simulated by models (Wang et al., 2020), although the observations suggest uncertainties in its sea-air exchanges, continental emissions and photochemical production and photodissociation rates (Fischer et al., 2012; Wang et al., 2020). Formic and acetic acid make up more than half of rain acidity in the remote atmosphere (Keene and Galloway, 1984), but their global budget remains poorly understood, and the significant discrepancies between modelled and measured distributions indicate large missing sources of both species in both polluted and remote regions (e.g. Paulot et al., 2011; Stavrakou et al., 2012; Millet et al., 2015; Khan et al., 2018). Although methanol is the most abundant tropospheric nonmethane compound, there are still uncertainties in its source apportionment (e.g. Jacob et al., 2005). Biogenic emissions are the largest source of methanol over continental areas (Stavrakou et al., 2011), and secondary photochemical sources appear to be the main contribution to methanol abundances over remote oceanic areas  (Bates et al., 2021). Finally, although methyl ethyl ketone (MEK) is much less abundant than acetone, it is also more reactive (Brewer et al., 2020). Besides diverse sources including photochemical production and anthropogenic emissions, there is evidence of significant biogenic and oceanic sources of MEK that can affect its overall abundance, especially in more remote locations (Yáñez-Serrano et al., 2016; Brewer et al., 2020). Recently, the deposition of isoprene oxidation products on vegetation and subsequent conversion to MEK and other products has been proposed to be the largest contribution to the MEK budget at the global scale (Canaval et al., 2020), although this estimate relies on relatively few measurements.

In this study, we confront multiple chemical observational datasets from the remote island of Réunion, in the southern Indian Ocean, with regional atmospheric composition simulations using the high-resolution Weather Research and Forecasting model coupled with chemistry (WRF-Chem). More specifically, we make use of Proton Transfer Reaction - Mass Spectrometry (PTR-MS) measurements of VOC and OVOC concentrations (Verreyken et al., 2021) performed at the high-altitude site of Maïdo Observatory (21.1° S, 55.4° E, 2160m a.s.l.). Despite its small size, Réunion Island has significant diversity in emission sources, and it undergoes the influence of oceanic and continental emissions (Baray et al., 2013). For these reasons, Réunion Island is an area of continuous interest, where long-term observations as well as dedicated measurement campaigns were conducted and used to validate large-scale models (e.g. Vigouroux et al., 2012; Callewaert et al., 2022) and analyze factors influencing local atmospheric composition  (e.g. Rocco et al., 2020; Rocco et al., 2022; Verreyken et al., 2020; Verreyken et al., 2021; Dominutti et al., 2022; Duflot et al., 2022). Due to these multiple influences and to the pronounced orography of the island, atmospheric chemistry modelling over Réunion is especially challenging. In



particular, the important role and strong spatial heterogeneity of anthropogenic and biogenic emissions (Verreyken et al., 2022) needs to be considered. Equally, the steep topography of the island makes high resolution indispensable in order to forecast the local circulation patterns (El Gdachi et al., 2024). In this work, the WRF-Chem model is therefore enhanced with high-resolution (1 km$^2$) emission datasets for anthropogenic emissions and biogenic isoprene emissions. Furthermore, the model is evaluated and further refined based on comparisons with meteorological and air quality in situ data, with

Fourier Transform Infrared Spectroscopy (FTIR) column measurements at Maïdo and with spaceborne (TROPOspheric Monitoring Instrument, TROPOMI) observations. Given the importance of meteorology for the simulation of transport, biogenic emissions and photochemistry, the model is evaluated against meteorological observations at Maïdo. FTIR and PTR-MS observations of long-lived compounds are particularly useful to evaluate the background atmospheric composition and to constrain the lateral boundary conditions of the regional model. TROPOMI column observations and air quality

measurements are essential to test and constrain the emissions, in particular for NOx. The PTR-MS dataset at Maïdo is expected not only to validate the model and better constrain the local emissions of important (O)VOCs, but also to identify potential shortcomings that should be considered in future studies of atmospheric composition in environments similar to Réunion Island.

The set-up and configuration of the WRF-Chem model, the chemical mechanism, the initial and boundary conditions and the

emissions used in the simulations presented in this study are described in Sect. 2.1-2.3. Sect. 2.4 presents the observational datasets used to evaluate the model, including meteorological observations, surface chemical concentrations data (Sect. 2.4.1), the PTR-MS dataset of VOC and OVOC concentrations at Maïdo (Sect. 2.4.2), Fourier-Transform infrared spectroscopy (FTIR) column measurements at Maïdo (Sect. 2.4.3) and finally, spaceborne columns from TROPOMI (Sect. 2.4.4). Sect. 3 evaluates the model performance relative to the various measurement datasets. The results are recapitulated

before concluding remarks in Sect. 4. Complementary figures and statistics can be found in the Supplement.

## 2. Methodology

### 2.1 Simulation area

Réunion is a French overseas island located in the Indian Ocean, ~700 km off the coast of Madagascar (Fig. 1). The island covers an area of 2512km$^2$ (63 km long and 45 km wide), and roughly spans from -21.39° S to -20.87° S, and 55.22°E to

55.84°E. Réunion is a volcanic island with mountainous orography, with a maximum altitude of 3070m above sea level (Piton de la Fournaise). The region is largely covered by native vegetation (100 000 ha) and is mostly free of strong anthropogenic emission sources (Duflot et al., 2019). The main anthropogenic emission sources are fossil fuel combustion in industrial and transport sectors (responsible for 87.1% of energy generation, Praene et al., 2012). The three principal industrial emission hotspots are due to biomass power plants (Le Gol and Bois-Rouge) and a diesel powerplant (Le Port)

(Fig. 2).





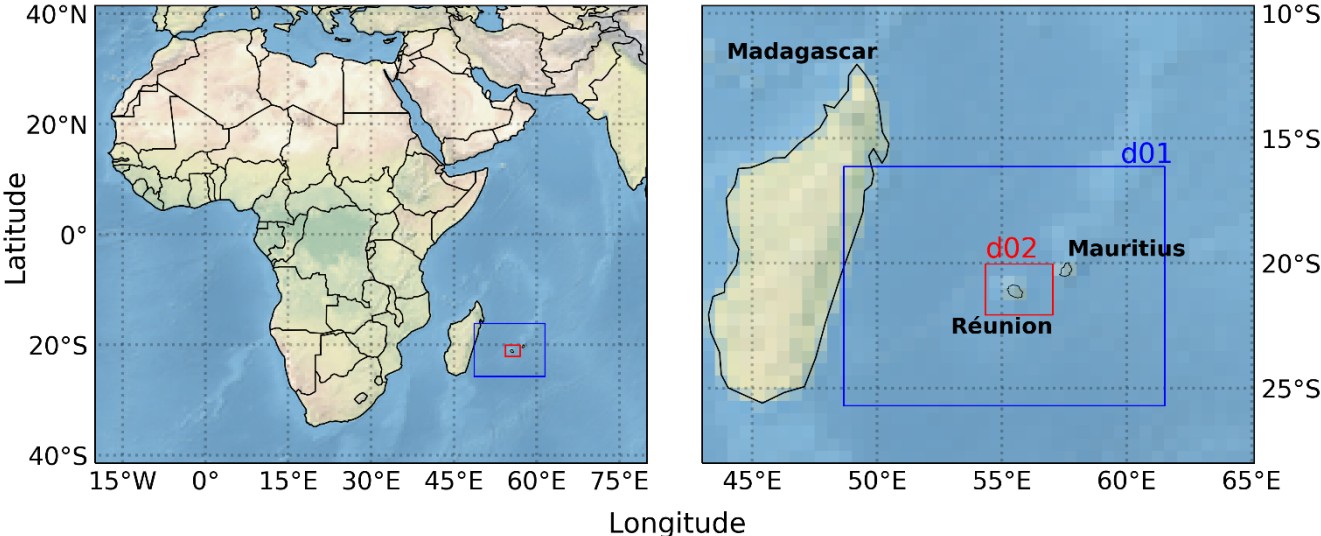

**Figure 1. Map of Africa and surrounding Indian Ocean indicating two model domains in blue and red (with 12.5 and 2.5 km horizontal resolution, respectively).**

Réunion Island has a large concentration of endemic species (Myers et al., 2000). The island is largely dominated by
vegetation and the plant species distribution changes with altitude (Fig. 2 in Strasberg et al., 2005; Foucart et al., 2018; Duflot et al., 2019). Over 80% of the human population is concentrated in the coastal regions.

## 2.2 WRF-Chem

The Weather and Research Forecasting model coupled with chemistry is used to calculate meteorological and chemical atmospheric processes (WRF-Chem; Grell et al., 2005). Model version 4.1.2 of WRF-Chem was used in conjunction with its
preprocessing system, WPS (WRF Preprocessing System) version 4.1.

### 2.2.1 Model configuration

To minimize computational demand, the simulations were conducted at 12.5 and 2.5 km horizontal resolution in the parent and nested domains, denoted d01 and d02, respectively (Fig. 1). Note that a 2 km horizontal resolution was found appropriate for simulating FTIR and in situ observations in a model study of greenhouse gases above Réunion, also using
WRF-Chem but with the chemistry turned off (Callewaert et al., 2022). The projection is set to Mercator, as is the recommended set-up for low-latitude simulations close to the Equator. Thirty-day simulations were conducted for January and July 2019, each starting on the first day of the month at 00:00 UT. The local time at Réunion is UT + 4h. January and July correspond to summer and winter in the southern hemisphere, although the difference in meteorology between the two seasons is relatively small under the tropical climate of the island. January and July in 2019 were relatively free of events
interfering with data collection (weather, volcanic activity, maintenance, etc.; see Table 1 in Verreyken et al., 2021). Lightning NOx emissions are ignored in the reference model simulation, but are included in a sensitivity run, using the





updated Price and Rind parameterization scheme based on cloud-top height (Price and Rind, 1992; Wong et al., 2013). Since
Barten et al. (2020) noted that the standard WRF-Chem settings lead to a large overestimation of lightning emissions, we
also downscale the number of flashes (adopted flash rate factor of 0.1 and 0.02 for d01 and d02, respectively, to account for
the difference in resolution) and the production of NO per lightning strike is set to 250 moles, instead of 500 in the standard
setting (Barten et al., 2020).

Simulations were conducted using Silicon Graphics (SGI) high-performance computing (HPC), equipped with an Intel
processor, using 72 cores (accounting for 5 h walltime for a two-day simulation in a 2-domain set-up). The simulations were
conducted sequentially in 2-day intervals, whereby the initial chemical conditions of each run are obtained from the previous
run, except in the case of the first day of the month. The meteorology is re-initialized at the start of each 2 day simulation.
The physical parameterizations incorporated in the simulations are listed in Table 1.

**Table 1. List of WRF-Chem physical parameterizations adopted for this study.**

| Model component | Name | Reference |
| --- | --- | --- |
| Microphysics | Morrison 2-moment scheme | Morrison et al. (2009) |
| Longwave radiation | RRTMG | Iacono et al. (2008) |
| Shortwave radiation | RRTMG | Iacono et al. (2008) |
| Planetary boundary layer | Shin-Hong scale-aware scheme | Shin and Hong (2015) |
| Surface layer | Revised MM5 scheme | Jiménez et al. (2012) |
| Land surface | Unified Noah land surface model | Tewari et al. (2004) |
| Cumulus parameterization | Grell 3D ensemble scheme | Grell (1993) Grell and Dévényi (2002) |
| Urban surface | Single-layer urban canopy model | Chen et al. (2011) |



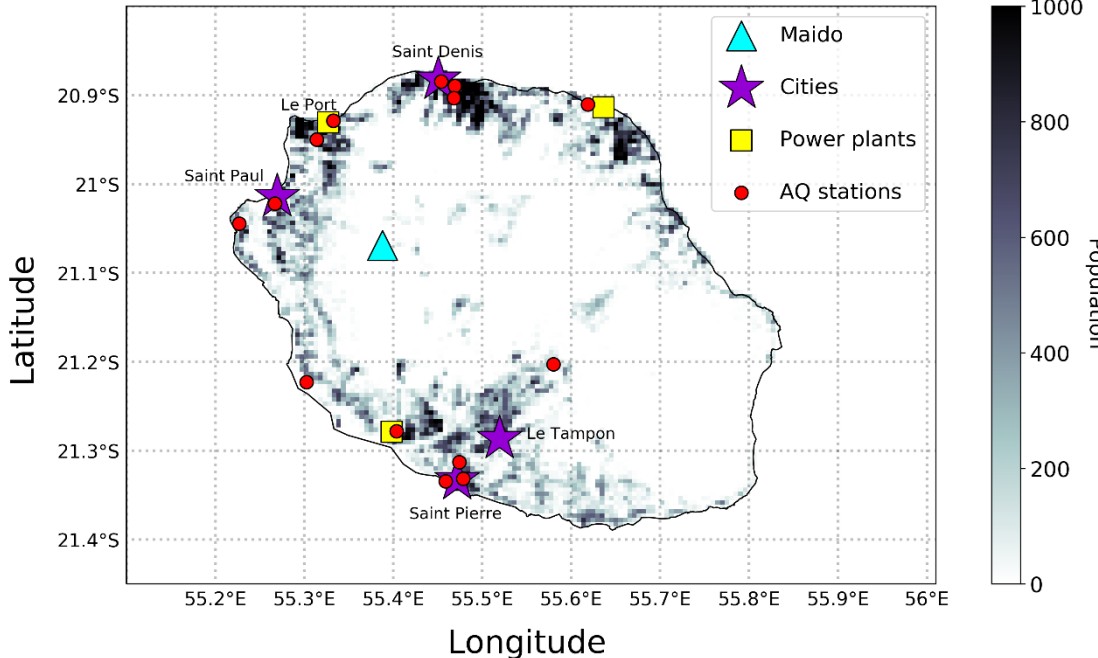

**Figure 2. Map of Réunion Island with population density at 500 m x 500 m resolution. Key locations are denoted on the plot; cities (with 50,000+ population, based on the 2020 Population Document, https://www.insee.fr/fr/statistiques/fichier/4265439/dep974.pdf, last access: 29 November 2023) and the principal power plants. The blue triangle, representing the Maïdo Observatory, is the location of the PTR-MS and FTIR instruments. The locations of the air quality monitoring sites are represented by the red circles.**

The land use dataset utilized in WRF is provided by MODIS 21 class (Friedl et al., 2002; Hulley et al., 2016), and has a resolution of 30 seconds (roughly 1km). The distribution of the MODIS land use index at the resolution of the nested domain is displayed on Fig. 3.

The dominant surface type at Maïdo is Woody Savannas (8), which is characterized by a tree cover between 30 and 60% and a tree canopy higher than 2 meters. There is spatial variability in the pixels adjacent to Maïdo, ranging from Evergreen Broadleaf Forests (2), Savannas (9) and Grasslands (10), each with different specifications regarding major vegetation type and tree cover percentage. Note that the MEGAN v2.04 module utilized in WRF-Chem recognizes only four vegetation types: broadleaf trees (BT), needle leaf tree (NT), shrub and brush (SB), and herbs, crops and grasses (HB). Their spatial distribution is provided by an independent dataset developed specifically for Réunion Island (see further below).



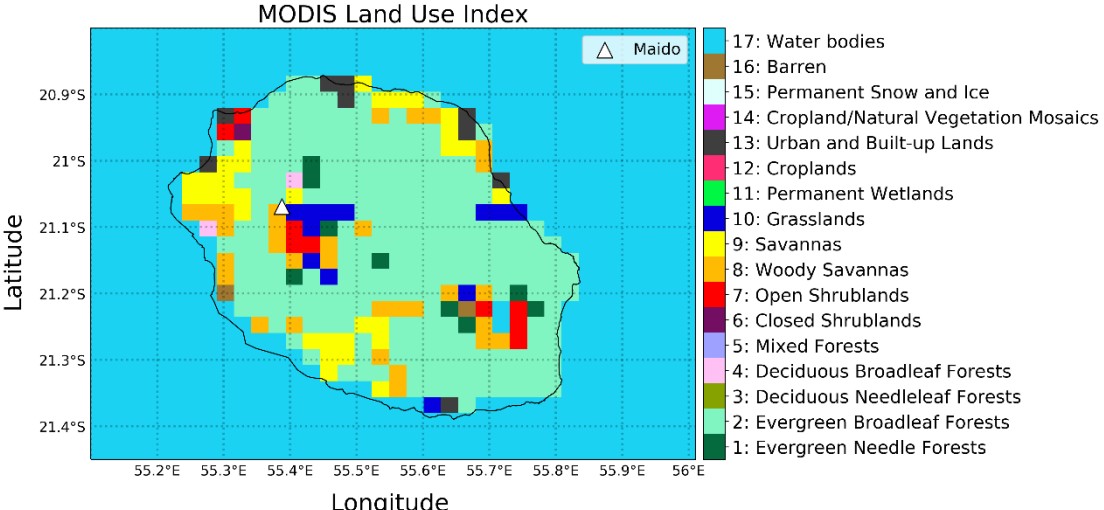

**Figure 3. Land use index used for defining surface features in WRF, following the classification of the MODIS Land Cover Type Product. The resolution shown is 2.5 x 2.5 km$^2$ (d02).**

### 2.2.2 Chemical mechanism

The reference gas-phase mechanism used in the model simulations is the Model for Ozone and Related chemical Tracers, version 4 (MOZART-4) mechanism (Emmons et al., 2010), with the Kinetic PreProcessor (KPP) (Damian et al., 2002). Several updates to the mechanism were tested and implemented, as described further below. Aerosol chemistry is simulated using the Global Ozone Chemistry Aerosol Radiation and Transport model (GOCART) (Chin et al., 2002). The MOZART-4 mechanism was chosen in lieu of its more updated counterpart, MOZART-T1 (Emmons et al., 2020) for computational reasons, despite the improvements of the new version, namely on isoprene, monoterpene and aromatic chemistry. Nevertheless, the isoprene chemistry of the MOZART-4 mechanism used in the model has been updated to reflect recent mechanistic updates regarding OH-recycling and formation of MVK/MACR (Sect. 2.2.3).

### 2.2.3 Mechanistic updates

#### 2.2.3.1 Isoprene

The MOZART-4 isoprene mechanism lacks the OH recycling mechanisms which were shown to occur through the reactions of isoprene hydroxy hydroperoxides (Paulot et al., 2009) and the unimolecular reactions of isoprene peroxy radicals (Peeters et al., 2009, 2014; Wennberg et al., 2018; Müller et al., 2019). The reactions of isoprene peroxy radicals (ISOPO2) and isoprene hydroxy hydroperoxides (ISOPOOH) in MOZART-4 are amended based on those studies and adjusted in order to match the results of a more up-to-date mechanism (MAGRITTEv1.1, Müller et al., 2019) through box model comparisons at both high NOx (1 ppb NOx) and low NOx (0.1 ppb NOx). More specifically, as seen in Table 2, the yields of MVK and MACR in the ISOPO2 reaction with NO are increased at the detriment of the hydroxy aldehydes (HYDRALD), and OH is formed in reactions that play a dominant role at low NOx (ISOPO2+HO$_2$ and ISOPOOH+OH), in order to mimic OH





recycling processes involving compounds that are missing from the mechanism. Note that the mechanistic changes were chosen to avoid the introduction of additional species to keep a similar overall computational cost.

**2.2.3.2 MEK oxidation**

The mechanism of methyl ethyl ketone (MEK) is revised with the aim to avoid overestimation of the photochemical
production of acetaldehyde through the oxidation of MEK and higher alkanes, which are MEK precursors. The oxidation of MEK by OH forms an intermediate radical product (MEKO2 in MOZART-4) which leads to the formation of acetaldehyde in the presence of NO, with a unit yield in MOZART-4 (reaction MO1 in Table 2). MEKO2 actually represents a lumping of three isomers denoted MEKAO2, MEKBO2 and MEKCO2 in the comprehensive Master Chemical Mechanism (MCM) v3.3.1 (Saunders et al., 2003; https://mcm.york.ac.uk/MCM/). Only one of the three radicals leads to acetaldehyde in the
presence of NO according to MCM, and the overall chemistry can be summed up as follows

$$\text{MEK} + \text{OH} \rightarrow 0.459 \text{ MEKAO2} + 0.462 \text{ MEKBO2} + 0.079 \text{ MEKCO2} \tag{r1}$$

$$\text{MEKAO2} + \text{NO} \rightarrow 0.24 \text{ RO}_2 + 0.24 \text{ HO}_2 + 0.76 \text{ HCHO} + 0.76 \text{ EO2} + \text{NO}_2 \tag{r2}$$

$$\text{MEKBO2} + \text{NO} \rightarrow \text{CH}_3\text{CO}_3 + \text{CH}_3\text{CHO} + \text{NO}_2 \tag{r3}$$

$$\text{MEKCO2} + \text{NO} \rightarrow \text{HCHO} + \text{C}_2\text{H}_5\text{O}_2 + \text{NO}_2 \tag{r4}$$

where RO2 denotes the acetonyl peroxy radical, $\text{CH}_3\text{C(O)CH}_2\text{O}_2$, and EO2 denotes $\text{HOCH}_2\text{CH}_2\text{O}_2$. Note that intermediary MCM reactions leading to the final products above were skipped as they involved compounds that are not defined in the MOZART-4 mechanism. These compounds were assumed to react rapidly according to their major sink reaction in presence of NO. Combining the equations above leads to the updated MO1 reaction shown in Table 2. The oxidation of MEK at low NO is oversimplified in the MOZART-4 mechanism and leads to the same final products as at high NO. However, the
reaction of the peroxy radicals MEKAO2, MEKBO2 and MEKCO2 with $\text{HO}_2$ form ketohydroperoxides that are expected to photolyse rapidly, leading in part to very different products (enols) than in the high NO case (Liu et al., 2018).

**Table 2. List of mechanistic changes adopted in the MOZART-4 mechanism.**

|  | Isoprene oxidation | |
|---|---|---|
|  | In MOZART-4 | In this work |
| IO1 | ISOPO2 + NO → .08 ONITR + .92 NO2 + .23 MACR + .32 MVK + .37 HYDRALD + .55 CH2O + HO2<br>Rate = 4.4E-12 exp(180/T) | ISOPO2 + NO → .10 ONITR + .9 NO2 + .30 MACR + .53 MVK + .07 HYDRALD + .83 CH2O + .92 HO2<br>Rate = 4.4E-12 exp(180/T) |
| IO2 | ISOPO2 + NO3 → HO2 + NO2 + .6 CH2O + .25 MACR + .35 MVK + .4 HYDRALD<br>Rate = 2.4E-12 | ISOPO2 + NO3 → HO2 + NO2 + .93 CH2O + .33 MACR + .59 MVK + .08 HYDRALD<br>Rate = 2.4E-12 |





| IO3 | ISOPO2 + HO2 → ISOPOOH  Rate = 8E-3 exp(700/T) | ISOPO2 + HO2 → ISOPOOH + OH  Rate = 8E-3 exp(700/T) |
|---|---|---|
| IO4 | ISOPOOH + OH → .5 XO2 + .5 ISOPO2  Rate = 1.52E-11 exp(200/T) | ISOPOOH + OH → .8 XO2 + .1 ISOPO2 + 1.7 OH + .2 CH3COCHO  Rate = 1.1E-10 |
| IO5 | ONITR + OH → HYDRALD + HO2 +.4 NO2  Rate = 4.5E-11 | ONITR + OH → HYDRALD + HO2 + .4 NO2  Rate = 3E-11 |
|  | MEK oxidation | |
| MO1 | MEKO2 + NO → CH3CO3 + CH3CHO + NO2  Rate = 4.2E-12 exp(180/T) | MEKO2 + NO → 0.11 RO2 + 0.11 HO2 + 0.428 HCHO + 0.349 EO2 + 0.462 CH3CO3 + 0.462 CH3CHO + 0.079 C2H5O2 + NO2  Rate = 4.2E-12 exp(180/T) |

### 2.2.4 Initial and boundary conditions

The model is initialized at the start of each run using input data from the Community Atmosphere Model with Chemistry (CAM-Chem; (Lamarque et al., 2012; Emmons et al., 2020)), which is a global model utilizing the MOZART-4 mechanism and providing output at 0.9° x 1.25° resolution. This allows for a direct representation of the initial and boundary conditions in the run. For species where the global Copernicus Atmosphere Monitoring Service (CAMS) reanalysis data (Inness et al., 2019) is available (NOx, CO, O₃, HNO₃, H₂O₂, HCHO, peroxyacetyl nitrate or PAN, C₂H₆ and C₃H₈), we replace the CAM-

Chem initial conditions with these higher resolution data (0.75° x 0.75°). Both inputs were taken at 6 hour intervals. Adjustments to these initial and boundary conditions were made based on comparisons with ground-based and FTIR measurements, which are detailed in Sect. 3.4. More specifically, the concentrations of methanol and PAN were multiplied by 0.6. Toluene, MEK and BIGALK were multiplied by 0.4 in both seasons, while the same factor was only applied to acetone in July. Ethane was increased by a factor of 1.6 in July and left unchanged in January.

### 2.3 Emissions

The source apportionment of the main gaseous pollutants and VOC precursors is summarized in Fig. 4, which provides the total emission for every species over the island (average of January and July). In addition, Fig. 5 displays the emission



distribution of the four most emitted VOCs over the island, namely isoprene, methanol, monoterpenes and acetaldehyde, for the month of January. The assumptions and datasets used to derive these emissions are given in the following subsections.

### 2.3.1 Anthropogenic Emissions

The horizontal resolution of global anthropogenic emission inventories such as EDGAR (0.1° x 0.1°) (Crippa et al., 2022) is too coarse for air quality simulations over Réunion Island. The low-resolution dataset cannot accurately represent the transition from highly polluted areas (e.g. cities) to remote zones such as Maïdo. We utilized a 1 x 1 km$^2$ emission inventory estimated for NOx, SO$_2$, CO, and NMVOCs based on data from the local air quality agency (Atmo-Réunion). The NMVOC species used in this study are listed in Table 3 below. The NMVOC speciation follows the Regional Atmospheric Chemistry Mechanism, Version 2 (RACM2) (Goliff et al., 2013). Traffic emissions were provided by Atmo-Réunion for the main roads of the island. Industrial emissions, mostly cane sugar refining, rum distillation and diesel-electric power production, were estimated as point sources. Agricultural emissions are distributed in cultivation areas.

The Atmo-Réunion emissions are complemented with the remaining sectors (namely shipping, residential burning, non-road transport, solvent processing, and waste management) from the EDGAR inventory, as well with species not included in the high-resolution inventory (ammonia, black and organic carbon and particulate matter). EDGARv6.1 (Crippa et al., 2022) was used for the trace gases and particulate matter, and EDGAR-HTAP v4.3.2 (Crippa et al., 2018; Huang et al., 2017) was used for the non-methane VOCs (NMVOCs). Those inventories are also used for all species and sectors in the rest of the model domain. Emissions from EDGARv6.1 reflect values from 2018, while emissions from EDGAR-HTAP v4.3.2 are based on values from 2012.

Temporal variations in the emissions are applied in accordance with Poraicu et al. (2023), based on EDGAR-specific temporal profiles (Crippa et al., 2020). Réunion being a French territory, the temporal profile follows the French specifications, which is defined based on French mainland regions. The temporal variation for some activities might not be representative of Réunion Island. For example, the seasonal temperature variation in mainland France is higher than in the tropics, with cold winters and hot summers, which determines a different residential heating behavior. The seasonal dependence of residential sector emissions was therefore omitted.

**Table 3: MOZART-4 VOC precursors and correspondence with the species classes of the Atmo-Réunion and EDGAR-HTAP V4.3.2 inventories.**

| MOZART-4 | Atmo-Réunion inventory (RACM2 compound classes, Goliff et al., 2013) | EDGAR-HTAP V4.3.2 VOC classes (Huang et al., 2017) |
|----------|-----------------------------------------------------------------------|-----------------------------------------------------|
| BIGALK | 0.6 * HC3 + HC5 + HC8 | voc4 + voc5 + voc6 + voc18 + voc19 |
| BIGENE | OLI + 0.5 * OLT | voc12 |



| $C_2H_4$ | ETE | voc7 |
|---|---|---|
| $C_2H_5OH$ | 0.85 * ROH | 0.85*voc1 |
| $C_2H_6$ | ETH | voc2 |
| $C_3H_6$ | 0.5 * OLT | voc8 |
| $C_3H_8$ | 0.4 * HC3 | voc3 |
| $CH_2O$ | HCHO | voc21 |
| $CH_3CHO$ | ACD | voc22 |
| $CH_3COCH_3$ | 0.253 * KET | 0.2 * voc23 |
| $CH_3OH$ | 0.15 * ROH | 0.15 * voc1 |
| MEK | MEK | 0.8 * voc23 |
| TOLUENE | TOL + BEN + XYM + XYO + XYP | voc14 + voc15 + voc13 |
| ISOP | ISO | voc10 |




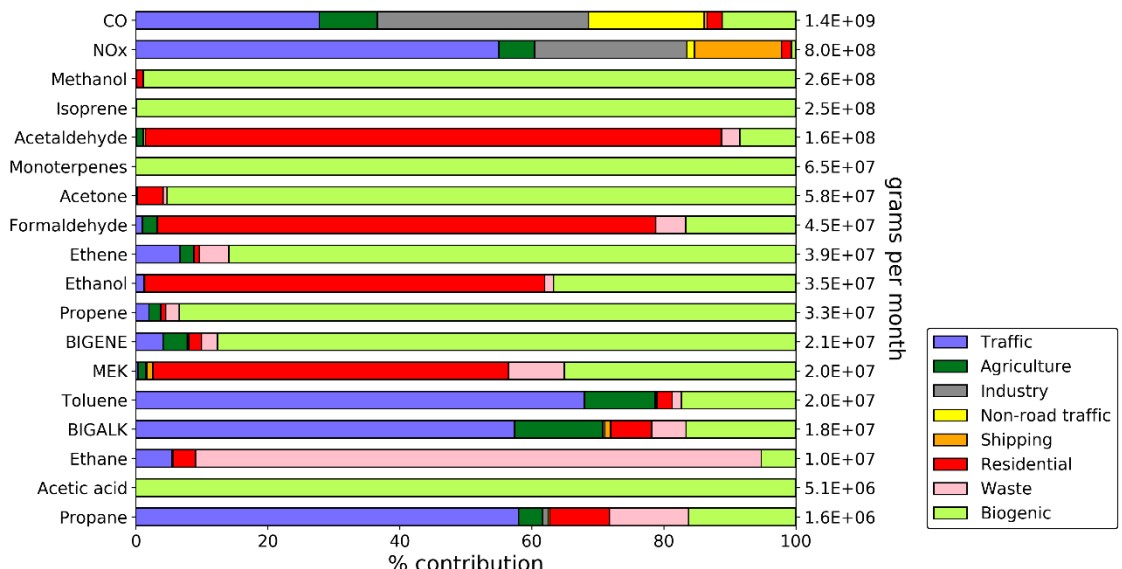

**Figure 4. Percentage contributions of different emission sources (anthropogenic and biogenic) to total emission of different chemical compounds considered in this work. The total, shown on the right-hand y-axis, is calculated from the average of the two months. Results shown for the optimal run (R0), after adjustment of the anthropogenic and biogenic sources described in text (Sects. 2.2, 2.3 and 3).**

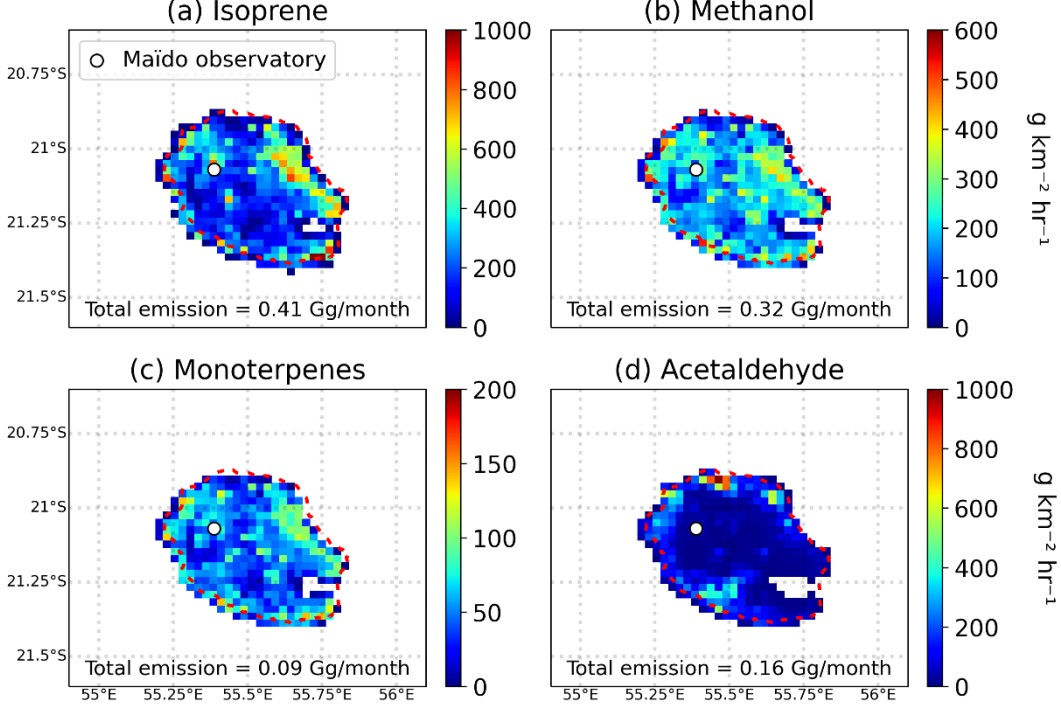

**Figure 5. Total emissions of isoprene, methanol, monoterpenes and acetaldehyde over the inner model domain at 2.5km resolution (d02), averaged over the month of January 2019. The white circle indicates the location of the Maïdo observatory.**



### 2.3.1.1 Anthropogenic emission adjustments

Given the coarse resolution (0.1°) of the EDGAR inventory used for residential emissions, those emissions were redistributed spatially on the model grid using a distribution of population density, specific to Réunion (https://public.opendatasoft.com/explore/dataset/population-francaise-par-departement-2018/table/?disjunctive.departement, last access: 8 December 2023). The population distribution is on a latitude-longitude grid at a resolution of 30m, and was regridded to the model resolution in the nested domain. In addition, since traffic activity occurs not only over highways but

also in cities and generally where people live, the traffic sector was redistributed by assuming that 70% of the emissions follow the population map, and 30% of the emissions are distributed on the principal highways defined in the Atmos-Réunion inventory. The resulting high-resolution distribution is compared with EDGAR on Fig. 6.

The NOx industrial emissions (primarily energy production) were redistributed based on the EDGARv6.1 emissions for this species and sector. Both the EDGARv6.1 dataset and TROPOMI $NO_2$ column data (see Sect. 3.5) suggest a maximum

emission in the northwestern region of the island, around the Port-Est power plant. Therefore, the total industrial NOx emissions were distributed amongst the 4 point source regions using percentage contributions based on the NOx industrial EDGARv6.1 emission dataset. EDGAR-HTAP v4.3.2 reports a total VOC emission three times higher than those from Atmo-Réunion, implying significantly lower VOC/NOx emission ratio in the latter inventory. Although both inventories exhibit similar spatial distributions, the Atmo-Réunion emissions are higher around the largest city (Saint-Denis) and lower

in the other industrialized areas.





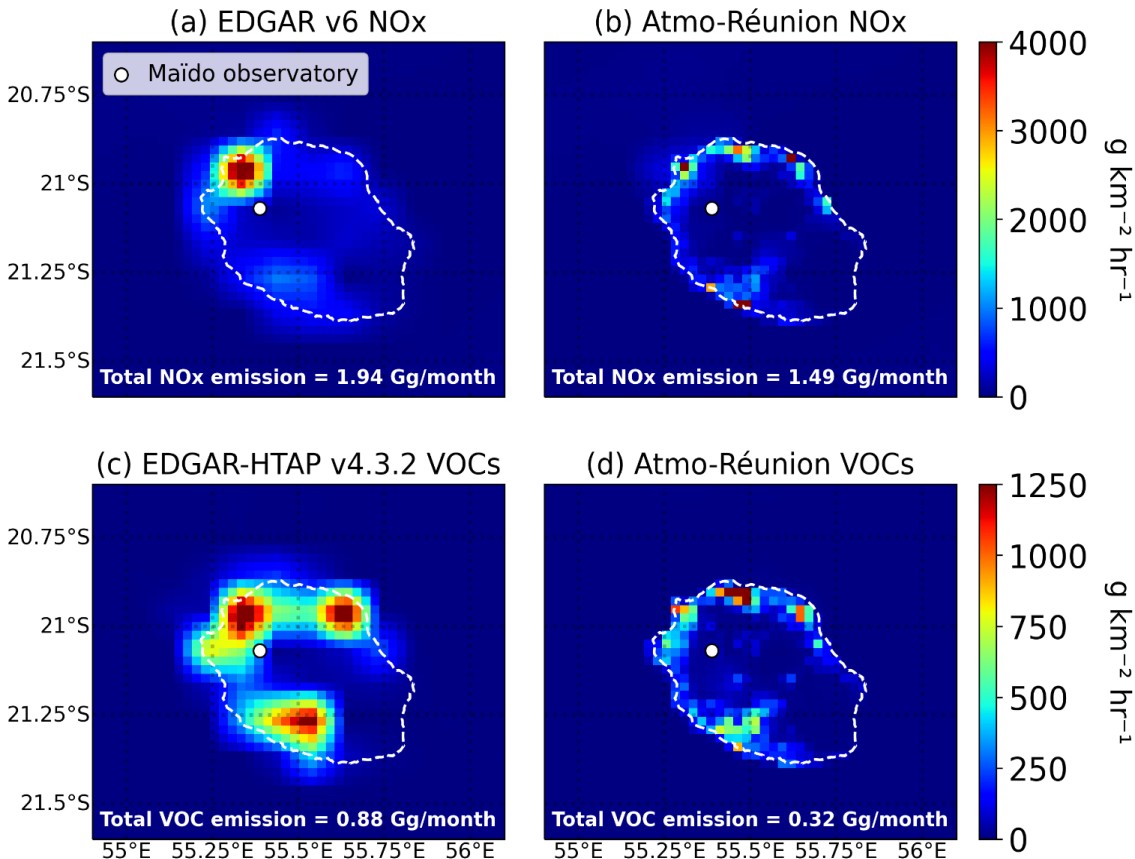

**Figure 6. Anthropogenic emissions of (a-b) NOx and (c-d) total VOC over the inner model domain at 2.5km resolution (d02), averaged over the month of January 2019. (a) and (c) display low-resolution EDGAR emissions, while (b) and (d) show the high-resolution Atmo-Réunion dataset, with adjustments described in this Section.**

### 2.3.2 Biogenic Emissions

Emissions from biogenic sources, mainly from vegetation, are calculated using the Model of Emissions of Gases and Aerosols from Nature (MEGAN) version 2.04. The emissions are calculated on-line (at the same time step as the model), based on the simulated meteorological fields and vegetation types defined by the land use map. This model utilizes 4 general vegetation types (broadleaf trees, needleleaf trees, shrubs, herbaceous land) together with standard emission factors and other parameters needed to estimate the emission at each timestep. MEGAN calculates the emissions for 20 compounds/compound classes, based on which the emission of 138 individual species can be estimated. The emissions of WRF-Chem species classes (e.g. monoterpenes) are obtained by lumping of those individual compounds. The biogenic emission for each class ($E$) is calculated using

$$E = \varepsilon \, \gamma \, \rho \, , \tag{1}$$





i.e., $E$ is calculated as a product of the emission factor at standard conditions ($\varepsilon$), the emission activity factor ($\gamma$) and a factor accounting for production and loss within the plant canopy ($\rho$). The latter factor is not considered here, i.e. $\rho$=1. The standard emission factor (in µg m$^{-2}$ hr$^{-1}$) is obtained from independent field and laboratory studies.

The dimensionless emission activity factor $\gamma$ accounts for the dependence on environmental conditions such as the leaf area index (LAI), the photosynthetic photon flux density, i.e. the amount of visible light at leaf level ($\gamma_P$), leaf temperature ($\gamma_T$),

leaf age ($\gamma_A$), soil moisture ($\gamma_{SM}$) and $CO_2$ inhibition ($\gamma_C$). The total gamma thus has a contribution from all sensitivities:

$$\gamma = \text{LAI } \gamma_P \, \gamma_T \, \gamma_A \, \gamma_{SM} \, \gamma_C \, C_{CE} \qquad (2)$$

$C_{CE}$ is a factor dependent on the specific canopy model used in the calculations. $C_{CE}$ is set such that $\gamma = 1$ at standard conditions.

The response to the photosynthetic photon flux density ($\gamma_P$) is calculated using

$$\gamma_P = (1 - LDF) + LDF \cdot C_P \left[ \frac{(\alpha \cdot PPFD)}{(1 + \alpha^2 \cdot PPFD^2)^{0.5}} \right], \qquad (3)$$

where LDF is the light-dependent fraction, defined in the model for each class of species. The last term in the equation represents the light-dependent activity factor. The photosynthetic photon flux density (PPFD) quantifies the amount of light received by leaf area. The sensitivity to leaf temperature is calculated using

$$\gamma_T = (1 - LDF) \, exp \left( \beta \left( T - T_s \right) \right) + LDF \left( E_{OPT} \left[ \frac{C_{T2} \cdot exp \, (C_{T1} x)}{C_{T2} - C_{T1} \cdot (1 - exp \, exp \, (C_{T2} x))} \right] \right), \qquad (4)$$

where $x$ is calculated as follows

$$x = \left( \frac{1}{T_{OPT}} - \frac{1}{T} \right) \cdot \frac{1}{0.00831}, \qquad (5)$$

$T$ is the leaf temperature (in K), and $T_S = 297$ K. $C_{T1}$, $C_{T2}$ and $\beta$ are empirically determined coefficients. $C_P$, $\alpha$, $T_{OPT}$ and $E_{OPT}$ are parameters dependent on the past conditions, namely the leaf level PPFD and temperature averaged over the past 24 and 240 h. For more detail on the MEGAN algorithm, see Guenther et al. (2006; 2012).

MEGAN relies on a 1 km resolution global map of plant functional types and leaf-area index. Since the standard database provided by NCAR (https://www.acom.ucar.edu/wrf-chem/download.shtml, last access: 1 December 2023) has zero values for all these variables over the Réunion area, we replaced it with a detailed cartography of plant functional types and isoprene standard emission factors (100m resolution) for Réunion Island constructed from an extensive survey of natural habitats (ZNIEFF, 2013) and phytosociology studies (Strasberg et al., 2005). This is completed by measurements of the leaf

area index (LAI) for representative plant species (Duflot et al., 2019). The resulting distributions of plant functional types, standard isoprene emission factor and leaf area index are shown on Fig. 7 at the model resolution (inner domain). As seen on this figure, broadleaf trees are by far the dominant PFT over the island. On average over the island (except bare soil), the isoprene emission factor is about 2900 µg m$^{-2}$ h$^{-1}$ (42.7 mol km$^{-2}$ h$^{-1}$). Around Maïdo observatory, the emission factor ranges between 3000 and 6000 µg m$^{-2}$ h$^{-1}$. Those values are much lower than the default values in the WRF-Chem model for the





dominant PFTs (13000 and 11000 µg m⁻² h⁻¹ for broadleaf trees and for shrubs, respectively). This illustrates the importance

of accounting for the local plant species distribution, when available.

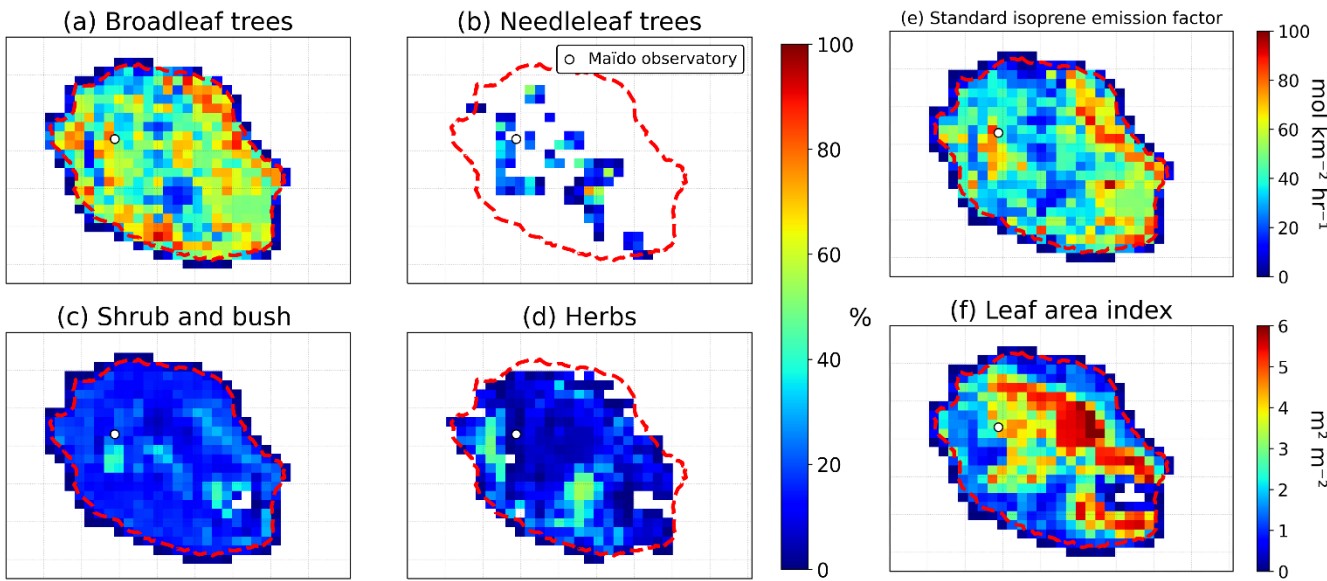

**Figure 7. (a-d) Plant functional type coverage (%) used to calculate biogenic emissions with MEGAN in WRF-Chem, (e) isoprene standard emission factor (mol km⁻² hr⁻¹) and (f) leaf area index at 2.5 km horizontal resolution. The white circle represents the location of the Maïdo Observatory.**


Except for isoprene, the emission parameters for each compound (or compound class) are assumed constant within each of

the 4 vegetation types, despite possibly important variations. In this work, the MEGAN parameters for several compounds

(methanol, monoterpenes, methyl ethyl ketone, acetaldehyde/ethanol) were updated based on previous work and on

comparisons with the PTR-MS measurements at Maïdo, which will be further discussed in in Sect. 3.3. More specifically,

the methanol emission factor was decreased from 800 to 400 µg m⁻² h⁻¹ for broadleaf trees and herbaceous vegetation, in

agreement with Stavrakou et al. (2011). The emission factors for monoterpenes were decreased by a factor of 5, and the

light-dependent fraction (LDF) was increased to 0.9 (from initial values ranging between 0.05 and 0.8 for different

monoterpenes). This high LDF value implies an emission algorithm similar to isoprene, with little emission at night. It might

reflect the dominance of broadleaf forests and the marginal extent of coniferous trees over the island (Fig. 7), suggesting a

minor role of temperature-controlled emission of VOC from storage pools (see e.g., Derstroff et al. 2017; Yáñez-Serrano et

al., 2015). The MEGAN2.1 emissions of methyl ethyl ketone (MEK) were enhanced, since recent studies showed an

important biogenic contribution to its budget (Yáñez-Serrano et al., 2016). Since this biogenic source is thought to result

from the partial conversion of isoprene oxidation products (methyl vinyl ketone, MVK, and isoprene-hydroxy-

hydroperoxides, ISOPOOH) deposited on plant leaves (Canaval et al., 2020), the biogenic source of MEK is assumed

proportional to that of isoprene. Although this assumption might underestimate MEK emission at night, since non-zero

concentrations of isoprene oxidation products might be sustained during nighttime (e.g. Langford et al., 2010), the error is





likely small since the dry deposition of those compounds is usually very slow at night (Nguyen et al., 2015). The scaling factor used for calculating the emission has been adjusted as discussed in Sect. 3.3.

### 2.3.3 Biomass burning emissions

The fire emissions from the Fire INventory from NCAR (FINN) emission inventory (Wiedinmyer et al., 2011) were zero over both domains in January and July 2019. FINN is a global, 1 km-resolution dataset providing daily estimates of biomass burning emissions. However, as for the biogenic emissions input files, the biomass burning emissions could be artificially missing in this inventory due to the small size of the island. In any case, there was no significant biomass burning activity detected at Maïdo in January and July 2019 (Verreyken et al., 2020). The fire emissions were therefore omitted in the

simulations.

### 2.4 Measurements

### 2.4.1 In situ surface chemical observations

Common air pollutants are measured hourly at 18 air quality stations around Réunion Island. These stations are operated by Atmo-Réunion (https://atmo-reunion.net/le-dispositif-de-surveillance, last access: 1 December 2023). The stations measuring

$NO_2$, $O_3$ or CO are shown on Fig. 2. In the comparison of modelled and observed $NO_2$ concentrations, a correction factor is applied to account for known interferences in the $NO_2$ measurement (Lamsal et al., 2008). These interferences are due to NOy reservoir compounds (such as $HNO_3$ and PAN) contributing to the measured signal and leading to higher $NO_2$ values than are actually present. The correction factor is calculated using $NO_2$, PAN and $HNO_3$ concentrations obtained from the model, and applied to the simulated $NO_2$, as detailed in Poraicu et al. (2023).

### 2.4.2 PTR-MS measurements

The PTR-MS instrument is deployed for the measurement of VOC species (Hewitt et al., 2002). The dataset (Verreyken et al., 2021) includes 13 molecules, or more precisely, 13 mass-to-charge ratios (m/z). The instrument (hs-PTR-MS, Ionicon Analytik GmbH, Austria) was located at the Maido Observatory high-altitude site (2160 a.s.l) (21.079°S, 55.384°E). PTR-MS employs a three-step process of ionization, separation and detection. Ambient air is introduced into a drift tube and

mixed with reagent ions ($H_3O^+$) to protonate the VOC compounds present in the sample. For some VOC compounds this protonation process is followed by (partial) fragmentation of the nascent excited protonated molecules. The ionized molecules are transported by means of an electric field towards a quadrupole for mass to charge separation and are detected by a secondary electron multiplier. The instrument was used in the multiple ion detection mode, resulting in a near-continuous data time series over a 2-year period (October 2017 - November 2019). The major isoprene oxidation products at

high NOx, methacrolein (MACR) and methylvinyl ketone (MVK), are measured conjointly due to their identical molar mass (70 g mol$^{-1}$) and molecular formula ($C_4H_6O$). Isoprene hydroxy hydroperoxides (ISOPOOH), which are major isoprene



oxidation products at low NOx, are known to react heterogeneously in the PTR-MS instrument and decompose to either MVK or MACR (+HCHO) (Rivera-Rios et al., 2014). Therefore the PTR-MS signal corresponding to MVK+MACR includes a contribution of ISOPOOH, which is significant at low NOx conditions. Although the exact conversion efficiency 390 of ISOPOOH into MVK+MACR is uncertain, a complete conversion is assumed here, and the corresponding signal will be referred to as Iox (isoprene oxidation product, MVK+MACR+ISOPOOH).

The PTR-MS signal for acetic acid (CH$_3$COOH) has interferences from other chemical species, namely glycolaldehyde, peroxyacetic acid, propanols and ethyl acetate (Baasandorj et al., 2015). The contribution of propanol is very low (<1%) and is not considered. Protonated glycolaldehyde has the same mass-to-charge ratio as protonated acetic acid (61 m/z), while 395 peroxyacetic acid and ethyl acetate were shown to fragment upon protonation with a resultant fragment ion (61 m/z) (Španel et al., 2003, Fortner et al., 2009). We therefore opt to compare the observations with the sum of the modelled concentrations of acetic acid, glycolaldehyde, and peroxyacetic acid, assuming a similar sensitivity for all three compounds. Ethyl acetate is not considered in the MOZART-4 mechanism although this compound has both direct emissions and formation from the chemical oxidation of ethers (Orlando and Tyndall, 2010 and references within). Formic acid (46 g mol$^{-1}$), also measured by 400 the PTR-MS, is not defined in the MOZART-4 mechanism, and is thus not considered further in this study. The signal at m/z 73 has a major contribution from methyl ethyl ketone (MEK) (Verreyken et al., 2021) but other compounds, namely butanal isomers and methylglyoxal (MGLY), also contribute (Yáñez-Serrano et al., 2016). The contribution of butanal is neglected here, being not calculated by the model. The estimated sensitivity of the instrument to MGLY being only a fraction (~0.7) of the sensitivity to MEK (Koss et al., 2018), the PTR-MS concentrations will be compared to the sum MEK + 0.7×MGLY 405 calculated using the model.

Benzene, toluene and xylenes are all measured at the site. However, in the MOZART-4 mechanism they are lumped and treated as toluene. Benzene, toluene and xylenes having different chemical lifetimes (Atkinson, 2000) and different emission distributions, the PTR-MS observations of aromatics cannot be evaluated with the model. In situ meteorological data (2-m temperature, relative humidity, wind speed and direction and solar radiation) is measured at and provided for the PTR-MS 410 site (Maïdo Observatory) at the same temporal resolution as the chemical concentrations (2.7 minutes).

### 2.4.3 FTIR measurements

Ground-based FTIR measurements have been performed at Réunion Island (Maïdo and Saint-Denis) since 2002, first on a campaign basis (Senten et al. 2008). Long-term measurements of VOCs and other compounds started in 2009 at Saint-Denis (Vigouroux et al., 2012). Since 2011, FTIR measurements in the prescribed NDACC (Network for the Detection of 415 Atmospheric Change) spectral range (600-4500 cm$^{-1}$) are performed at the Maïdo Observatory (Baray et al. 2013), using a Bruker 125 HR (since 2013).

The FTIR being a remote sensing technique which measures the absorption of solar light by atmospheric species along the line-of-sight (instrument-sun), the primary product of the FTIR retrievals is the total column of the absorbing gases. In addition, low resolution vertical profile information can also be derived by using the pressure and temperature dependence of





the line shape. The choice of spectral windows and spectroscopic parameters is optimized for each target species, and preferably within the whole NDACC FTIR community to ensure a consistency within the network. Table S1 summarizes the main retrieval settings for the species used in this paper.

The HCHO retrieval has been optimized recently at more than twenty FTIR stations (Vigouroux et al., 2018) and is now an official NDACC target species. Methanol is not an official NDACC species, and is measured at only a few sites (see e.g.
Wells et al., 2024) but not in a harmonized way. Ethane is an official NDACC species and has been harmonized within the network as described in Franco et al. (2015). At Maïdo, the humidity being important, the third window suggested for harmonization in Franco et al. (2015) is not used to avoid strong interference with water vapor lines. Carbon monoxide is also an NDACC target species, and as such the windows and spectroscopy are harmonized and found in the NDACC InfraRed Working Group (IRWG) documentation (https://www2.acom.ucar.edu/irwg). The current IRWG retrieval strategies
for official species are currently being reprocessed, with the aim of using improved settings. Ozone is the FTIR species with the most degrees of freedom for signal (DOFS = 4 to 5), allowing the retrieval of several independent partial columns: one in the troposphere and three in the stratosphere (Vigouroux et al., 2008).

Peroxyacetyl nitrate (PAN) is not an official target species. It has been measured at only a few stations (Mahieu et al., 2021) because its weak spectral signature makes it difficult to detect and retrieve. This is the first time that PAN time-series at the
Maïdo station are presented. From the two stronger PAN absorptions used in Mahieu et al. (2021), only the band at 1163 cm$^{-1}$ is useful at Maïdo, the other one having too low signal-to noise-ratio. As shown in Fig. S1, the spectral signature of PAN is much weaker than the other absorbing gases. Therefore, the interfering species must first be carefully taken into account. In addition to the gases shown in Fig. S1, HCFC-22 also interferes. We pre-retrieve HCFC-22 in a dedicated window (828.62-829.35 cm$^{-1}$), and then, for each individual spectrum, the retrieved profile of HCFC-22 is used as fixed values in the PAN
retrieval. For each spectrum, profile retrievals are also made for $H_2O$, HDO, $O_3$, $N_2O$, and CFC-12 and used as a priori values in the PAN retrievals, in which PAN and $H_2O$ profiles are retrieved, while the other species are scaled from their a priori pre-retrieved profiles (with the exception of HCHC-22 which is not scaled, but fixed).

**2.4.4 TROPOMI observations**

The TROPOspheric Monitoring Instrument (TROPOMI) is a spaceborne instrument aboard the European Space Agency
(ESA) Sentinel-5P (S5P) satellite (Veefkind et al., 2012), which monitors the global distribution of multiple air pollutants, among which nitrogen dioxide, formaldehyde, ozone and carbon monoxide. The S5P satellite is sun-synchronous and the retrieved data has a daily global coverage with (typically) one TROPOMI measurement at ~13:30 local time, with a spatial resolution of 7 x 3.5 km$^2$ (updated to 5.5 x 3.5 km$^2$ in August 2019). The main species of interest for this study are $NO_2$ and HCHO. The measurement of multiple species is possible due to the large spectral range of the TROPOMI spectrometer,
specifically ultraviolet (UV), visible (VIS), near-infrared (NIR) and shortwave infrared (SWIR) ranges (ranging from 267 to 2389 nm). The instrument is a push-broom spectrometer that scans the Earth while the satellite moves and measures the composition of the atmosphere using differential optical absorption spectroscopy (DOAS). The light travels from the Sun





through the atmosphere and is reflected back to space, where it is measured by the spectrometer. Molecules absorb photons at well-defined windows, depending on their molecular structure (primarily 405 - 465 nm for NO$_2$ and 328-359 nm for
HCHO). Comparison of the observed spectrum to a reference spectrum enables the retrieval of the slant columns through a fitting procedure involving multiple compounds that are active in the absorption bands of the species of interest. For both compounds, TROPOMI retrieves a slant column density (SCD) from the Level-1b radiance and irradiance spectra, that represents the total amount of compound present along the effective solar light path (van Geffen et al., 2020).

In the case of NO$_2$, the total SCD combines both tropospheric and stratospheric distributions. The tropospheric slant column
is obtained by subtracting the stratospheric contribution from the total SCD. This contribution is obtained from a global chemical transport model (TM5-MP, Williams et al., 2017). The tropospheric vertical column density (VCD) is obtained by dividing the slant column by an air mass factor (AMF) (Palmer et al., 2001) which is dependent on the vertical profile of the considered compound. AMFs are obtained from radiative transfer calculations and can introduce a large source of uncertainty in the VCD calculation (30-40%, Lorente et al., 2017), especially in the presence of clouds. The vertical
distributions of NO$_2$ and HCHO are taken from the global chemistry transport model TM5-MP (van Geffen et al., 2022a; Williams et al., 2017; De Smedt et al., 2018) at a resolution of 1° x 1°. In the HCHO VCD algorithm, a background correction is applied on a daily basis. Since the reference spectrum is obtained from Earth radiances in the equatorial Pacific, the slant column retrieved from the fit corresponds to an excess over the remote background, where the principal HCHO source is methane oxidation. The TM5-MP columns over the same region are therefore added to the vertical columns to
account for this background (De Smedt et al., 2021).

Due to its role for the calculation of the AMF, the choice of vertical profile has an impact on the VCD retrieval, and it is necessary to consider the vertical sensitivity of the TROPOMI instrument when comparing with other datasets. NO$_2$ VCD biases against independent measurement datasets decreased when the TM5-MP a priori was replaced with model profiles at higher resolution (e.g Tack et al., 2021; Judd et al., 2020; Douros et al., 2023) or with measured profiles (e.g. Dimitropoulou
et al., 2020). In our study, the averaging kernels of the satellite data are applied to the model profiles to calculate a "smoothed" vertical column for comparison with the satellite retrieved columns (e.g. Boersma et al., 2016). The averaging kernel represents the sensitivity of the measurement to the tracer concentrations at different altitudes, weighted by the assumed vertical profile of the tracer (Eskes and Boerma, 2003). It is provided alongside the measurement in the TROPOMI products.

We present comparisons between the reprocessed version (RPRO) of the NO$_2$ and HCHO retrievals from TROPOMI (v3.2) and simulated tropospheric columns. For both compounds, the averaging kernels are applied to the model profiles vertically interpolated to the TM5 vertical pressure grids. Both WRF-Chem and TM5 vertical pressure levels are calculated using hybrid sigma-pressure coordinates and the surface altitude using values from WRF-Chem and TM5, respectively. Quality filtering (QF) follows Algorithm Theoretical Basis Document (ATBD) recommendations (QF > 0.75 for NO$_2$ and QF > 0.5
for HCHO) (van Geffen et al., 2022b; De Smedt et al., 2022).





In addition, we applied the oversampling technique to the NO$_2$ and HCHO data in order to gain insight into the fine-resolution distribution of those compounds (e.g., de Foy et al., 2009). This technique consists in the long-term averaging of TROPOMI measurements on a very fine grid, taken here to be 0.01° × 0.01° (~1 × 1 km$^2$). The measurement from a given TROPOMI pixel is taken to apply to a circle defined by the center of the pixel and a radius of 3.5 km. In this way, each 0.01°
× 0.01° pixel accumulates >500 measurements over the considered time period (May 2018 - July 2022). This technique takes advantage of the variable offset of TROPOMI observations from day to day and achieves a high signal-to-noise ratio at high resolution but is not intended for direct comparison with the model, given the long-term averaging. It aims to present the average pollutant distribution at finer scales to inform about emission hotspots that are potentially lost when considering data over short periods of time.

**3. Results and discussion**

The optimal model set-up for this region was obtained through multiple sensitivity runs testing various changes to the emissions, lateral boundary conditions and chemical mechanism used in the model. In this section, the reference run (R0) represents the simulation set-up incorporating all model updates. The sections below highlight the impact of the updates through model comparisons with the observations. The list of sensitivity runs portrayed in this study is provided in Table 4.

**Table 4: Sensitivity runs conducted in this study, with the shorthand notation.**

| Shorthand | Description |
|---|---|
| R0 | Best run with following adjustments:<br>• downscaling of principal NOx power plant emission by a factor of 5 (Sects. 3.2 and 3.5)<br>• adjustment of lateral boundary conditions (Sect. 2.2.4)<br>• adjustment of biogenic VOC emissions (Sect. 2.3.2)<br>• updates to the MEK and isoprene chemistry within the MOZART-4 mechanism (Sect. 2.2.3) |
| S1 | As R0, without the downscaling of NOx power plant emission by a factor of 5 |
| S2 | As R0, without the adjustment of lateral boundary conditions |
| S3 | As R0, without the biogenic VOC emission adjustments |
| S4 | As R0, without updates to the MEK and isoprene chemistry within the MOZART-4 mechanism |
| S5 | As R0, with direct anthropogenic and biogenic emissions of acetaldehyde, acetone, MEK, formaldehyde and acetic acid set to zero |



| S6 | As R0, including lightning emissions (Sect. 2.2.1) |
|---|---|

### 3.1 Evaluation against meteorological measurements

The model is evaluated against meteorological measurements at the Maïdo site on Fig. 8. Surface temperature and solar radiation are both represented by the model in both January and July. The observed diurnal profile of temperature is well
reproduced, except for a low nighttime bias on many instances. On most days in both months, the midday modelled value matches the observation well. Exceptions occur, e.g. during the first days of January, when the model underestimates temperature and overestimates relative humidity and cloudiness. Errors in the WRF-Chem cloud parameterization were shown to impact the model performance for many physical and chemical processes (Zhao et al., 2012; Berg et al., 2015; Ryu et al., 2018). Furthermore, cloudy periods (when observed solar radiation is lower, e.g. on Jan. 8-9) are typically not well
represented as such. The higher solar radiation fluxes in January (maximum of about 1000 W m$^{-2}$ in January compared to 750 W m$^{-2}$ in July) might enhance evaporation of the ocean and other water bodies, leading to a slightly higher humidity in the warmer month, as indicated by both model and measurement data in Fig. 8. Nevertheless, the seasonal variation of meteorological parameters is well reproduced by the model. January is warmer (by 5.5 and 6.4 K according to the measurements and the model, respectively), more humid (by 14 and 11%) and has higher radiation fluxes (by 51 and 65 W
m$^{-2}$) than the month of July. On average, the wind speed is slightly overestimated in both seasons. It is generally close to the observations, except during several predicted high-wind episodes (e.g. on 7-9 January and 24 July) that are not present in the observations. The predominant wind direction is correctly predicted (mostly westerly winds during January, and an alternance of westerly, northerly and easterly winds during July), but the model often fails to reproduce the short-term variability of that direction. For example, the sporadic occurrence of easterly winds (270°) in January is missed by the
model, whereas sporadic southerly winds are simulated in January and July but are seldom observed. The observed circulation likely results from the competition between overflowing trade winds and meso-scale dynamics induced by anabatic thermal flow coupled with upslope transport to the Maïdo Observatory, a direct consequence of orography (Duflot et al., 2019; El Gdachi et al., 2024). This will directly affect transport patterns and model comparisons with chemical measurements over the site. Seasonal averages are given in Table S2.








**Figure 8.** Evolution of observed and modelled surface meteorology at Maïdo observatory for (a-e) January and (f-j) July. The observations are shown in black, while the cyan line represents the WRF-Chem results. The wind direction is represented in degrees, relative to north.






## 3.2 Evaluation against air quality station measurements

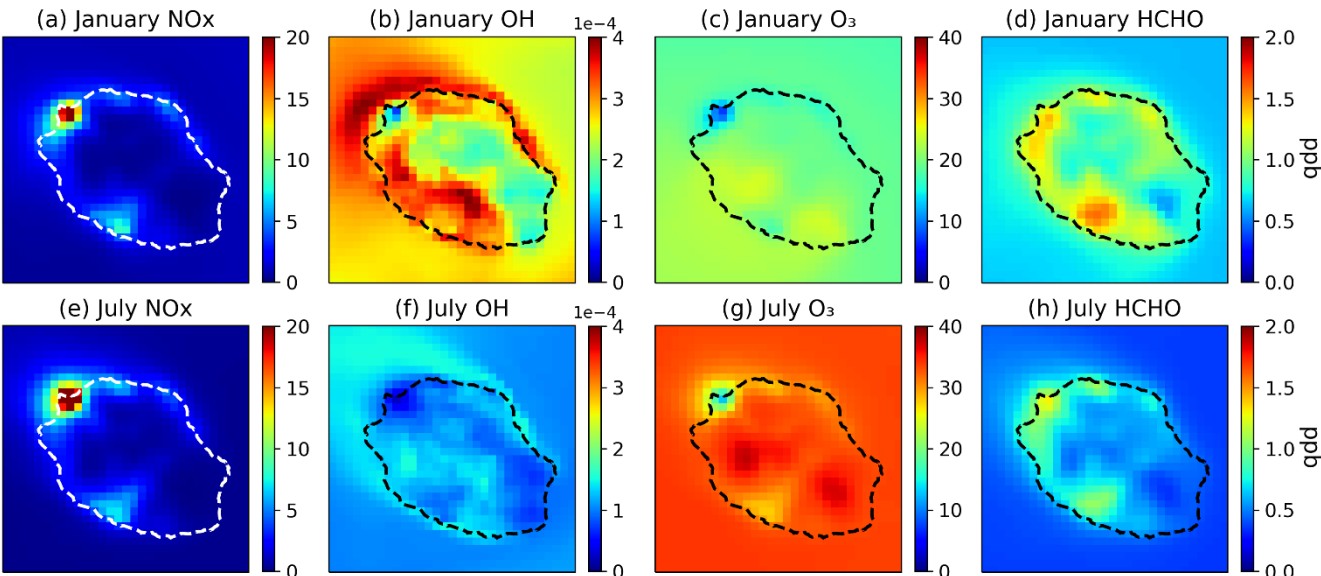

**Figure 9. Averaged modelled surface concentrations of NOx, OH, O₃ and HCHO (ppb) in January (top row) and July (bottom row), obtained from simulation R0.**

Figure 9 presents the monthly averaged distributions of key chemical compounds calculated by the model over Réunion Island, for January and July 2019. These distributions will be useful for the discussion of model comparisons with in situ and satellite observations (see next subsections). The strong heterogeneity of anthropogenic emissions and their high intensity over urban and industrial centers lead to strong contrasts between chemical regimes over different regions of the island. For example, the NOx mean surface concentration ranges between 0.12 ppb at the most remote part of the island to typically 5-

15 ppb over most urban/industrial areas, and over 30 ppb near the Le Port thermal plant. As seen on Fig. 9, other important compounds like OH and O₃ are strongly impacted by anthropogenic emissions. Over a large portion of the island (southwestern part as well as a small portion of the northern coast) but also over a large oceanic region surrounding almost the entire island (especially to the northwest due to the influence of Le Port emissions), OH levels are enhanced, primarily because NOx promotes the conversion of $HO_2$ to OH (Spivakovsky et al., 2000) through the reaction

$HO_2 + NO \rightarrow OH + NO_2$                                                           (r5)

As a result, OH increases as NOx increases when NO ranges between ca. 10 ppt and ca. 500 ppt (Logan et al., 1981). At very high NOx levels (> ~5 ppb NOx), OH decreases as NOx increases, mainly due to the sink of HOx (=OH+HO₂) due to the radical termination reaction

$OH + NO_2 \rightarrow HNO_3$                                                                       (r6)

which explains the depletion of OH levels over the main NOx hotspots. NOx has a relatively weak effect on the patterns of the ozone distribution, except for the clear titration effect near Le Port and the main cities, such as Saint Denis and Saint





Pierre. Formaldehyde is clearly strongly affected by anthropogenic activities, as seen from the high surface concentration levels around the NMVOC emission areas (Fig. 6). The distribution also partly reflects the complex impacts of NOx on OH, since the NMVOC oxidation processes leading to HCHO formation are mainly driven by their reaction with OH. Note that the NOx concentrations rapidly decrease with altitude. Therefore the OH depletion effect due to NOx seen on Fig. 9 becomes rapidly negligible at higher altitudes. Therefore, the main effect of anthropogenic NOx emissions on the oxidative conditions above the island is a strong enhancement of OH concentrations, except over localized hotspots, very close to the surface. NOx plays a major role in the OH budget, and inaccurate predictions of its abundances can affect the model comparisons for VOCs and their oxidation products, in particular at Maïdo. Besides the insights provided by TROPOMI $NO_2$ column data (see Sect. 3.5), network measurements of in situ concentrations of $NO_2$, NOx (=NO+$NO_2$) and $O_3$ are available mostly in polluted areas, in cities and in the vicinity of industrial sources (Fig. 2). Due to representativeness issues, caution is required when comparing model results with measurements often obtained very close to strong pollution sources. For this reason, model underestimation of NOx levels is to be expected at many stations. A useful indicator is the [NOx]/[$NO_2$] ratio. At photochemical steady state (PSS), it is given by

$$\left(\frac{[NOx]}{[NO_2]}\right)_{PSS} = 1 + \frac{J_{NO_2}}{k_1 \cdot [O_3] + \cdots}, \tag{6}$$

where $J_{NO_2}$ is the photolysis rate of $NO_2$, and $k_1$ (=1.95×10$^{-14}$ molec.$^{-1}$ cm$^3$ s$^{-1}$) is the rate of the reaction

$$O_3 + NO \rightarrow NO_2 + O_2 \tag{r7}$$

During the night, the expected value of the ratio is unity. However, close to a pollution source, photochemical steady state is not achieved, since directly emitted NO can travel over a few hundred meters within its chemical lifetime, of the order of several minutes for typical nighttime ozone levels (a few ppb, see Fig. 10). Furthermore, ozone is titrated to even lower levels in direct vicinity of strong sources. Given the model resolution and the distribution of NOx sources (Fig. 6), strong and systematic ozone titration occurs only in the region of Le Port (Figs. 2 and 9), with its strongly emitting power plants. Elsewhere, the nighttime [$NO_2$]/[NOx] ratio is below 1.1 in the model. In the measurements, however, the observed ratio is usually ~1.3, based on monthly averaged nighttime concentrations, and it even exceeds a factor 2 at several stations (Table S3), namely the stations 10-12 on the southern coast of the island (see Fig. S2 for station locations). These high values suggest the presence of very close NOx sources that are unresolved by the model.

Figure 10 presents a comparison of observed and modelled surface concentrations of NOx and $O_3$ for a 30-day simulation in January 2019. The stations were divided into two classes – the two stations near the Le Port power plants (LP) and the other stations, excluding the least representative stations, i.e. those for which the observed nighttime [NOx]/[$NO_2$] ratio exceeds a factor of 2 (Table S3).





**Figure 10. Evaluation of the average modelled surface $NO_2$, NOx and $O_3$ in January, from simulations S1 (blue) and R0 (red), against in situ network observations (dashed line) (a-b) in the region of Le Port (LP) and (c-e) at the other stations, for which the observed nighttime ratio [NOx]/[$NO_2$] < 2 (stations 3-9 of Table S3, see Fig. S2). The model $NO_2$ concentrations have been corrected to account for interferences in the measurements using the corresponding modelled concentrations of PAN and $HNO_3$ (Poraicu et al., 2023).**



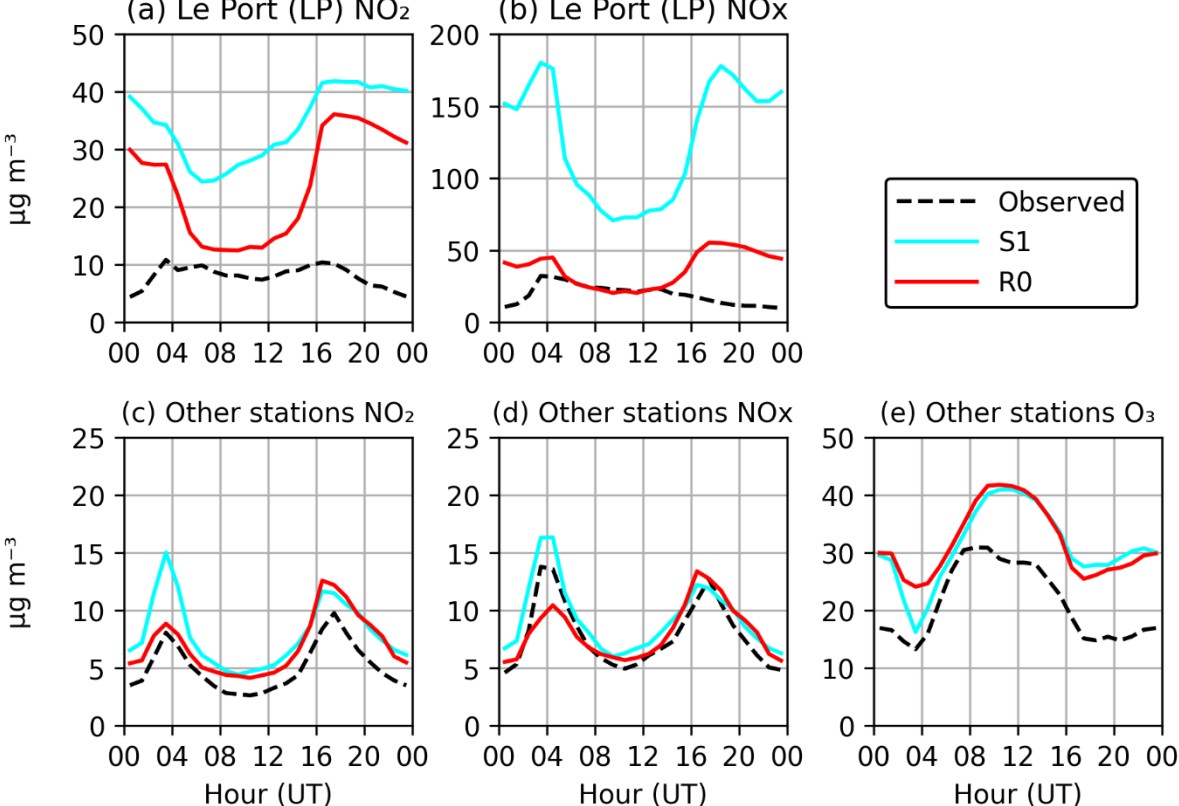

**Figure 11. Average diurnal profile of NO₂, NOx and O₃ in January, from simulations S1 (blue) and R0 (red), shown for the Le Port (LP) stations, and at the other stations for which the observed nighttime ratio [NOx]/[NO₂] < 2. Note that ozone measurements were not available at LP stations. The black dashed line represents the observations at the same stations.**

The LP stations (green dots on Fig. 2, Terrain de Sel and Centre Pénitentiaire) are strongly impacted by the NOx emissions due to the power plants located near Le Port, on the northeastern coast of the island (Fig. 9). However, the observed nighttime NOx concentrations are largely overestimated by the model at those stations. During daytime, a large overestimation is also found for simulation S1, which adopted the original estimation of the NOx emissions due to the Le Port power plants. On average over the 30 day period, the model overestimation for NOx (NO₂) against the LP measurements, a factor of 3 (4) during daytime in the S1 simulation, is strongly reduced (to a factor of 1-1.4) when the power plant emission is reduced by a factor of 5 (run R0) (Fig. 11). This finding suggests that this reduction of the power plant emission is justified, in agreement with the comparison of NO₂ columns with TROPOMI data (Sect. 3.5). The large model overestimation of nighttime NO₂ and NOx at the LP stations stands in contrast with the fairly good agreement at the other stations (see further below). A possible reason might be that all emissions are injected at surface level in the model, whereas the actual injection heights of power plant emissions might be above the (usually stable) nighttime boundary layer. The impact of the 5-fold reduction of NOx emissions is small (-25%) on the nighttime NO₂ levels at the two LP stations, despite their close proximity to the power plants. This is explained by the fact that NO (not NO₂) is emitted by the power



plants, and that a very large emission leads to $O_3$ titration through the reactions of NOx with $O_3$. The much stronger ozone

titration in the S1 run, compared to R0, explains that NO is much less rapidly converted to $NO_2$ in S1, compensating the higher emission. The titrating effect of the strong NOx emissions on the ozone distribution is clearly seen on Fig. 9, especially at and near Le Port and to a lesser extent at other industrialized centers.

By comparison, the effect of this emission reduction is small at the other stations, except during the morning traffic peak (4 UT i.e. 8 LT, Fig. 11). The model correlates very well with the observed $NO_2$ and NOx time series, although the model

performance is lower on the first and last days of the month, coinciding with unusually large errors in meteorological parameters at Maïdo (e.g. solar radiation and wind direction, see Fig. 8) . The average diurnal cycle of concentrations shows a good model agreement for $NO_2$ and NOx during night and day, except for a moderate overestimation of $NO_2$ (Figs. 10 and 11). This discrepancy between the model agreement for NOx and for $NO_2$ is likely due to the model overestimation of surface $O_3$, by more than 10 µg m$^{-3}$ on average, leading to an overestimated $[NO_2]/[NO]$ ratio (reaction r7). Although the S1

run achieves a better agreement with $O_3$ data at the morning peak hour, the performance of the two simulations is similar at other times of the day, and the shape of the diurnal profile is better represented by the R0 run. The reasons for the ozone overestimation (by ca. 12 µg m$^{-3}$ on average) are unclear, but could be due to the neglect of halogen chemistry in the WRF-Chem version used in this study. For example, recent studies indicated that the inclusion of halogen (Cl, Br, I) chemistry in a regional or global model might deplete near-surface ozone levels by a much as 7 ppb (~14 µg m$^{-3}$) in the marine tropical

troposphere (Badia et al., 2019; Caram et al., 2023). This reduction of ozone is partly due to a shortening of its lifetime, primarily attributed to iodine chemistry, and partly due to a reduction of ozone production, which is a consequence of the depleting effect of halogens on NOx levels (Caram et al., 2023; Iglesias-Suarez et al., 2020; Sherwen et al., 2016). Since this influence of halogens is ignored in our model, the good agreement of the model with in situ NOx measurements (Figs. 10 and 11) might mean that our NOx emissions are actually too low, although a definitive assessment is not possible at this

stage.

### 3.3 Comparisons with PTR-MS measurements

Figs. 12 and 13 show modelled VOC concentrations (from simulations R0, R2, S3, S4 and S5) against observations at the Maido observatory for the months of January and July 2019. The averaged diurnal cycles of observed and simulated concentrations are shown on Fig. 14. For most species, the general evolution is well represented by the model. Both the

observed and modelled diurnal cycles display a pronounced daytime maximum for almost all species. This general behavior reflects the dominance of daytime sources for most compounds but also the weaker influence of surface emissions during the night, when the station of Maïdo is frequently located in or near the free troposphere (Verreyken et al., 2021).

### 3.3.1 Formaldehyde

Both the model and the measurements display a pronounced diurnal cycle of formaldehyde. The first and last days of each

month show the best agreement with measurements, matching both day- and nighttime values closely. On the other days,





there is a consistent underestimation, especially at night, when it can reach a factor of two. This underestimation is unexpected, given the model overestimation of FTIR columns (Sect. 3.4) and the fact that nighttime observations at Maïdo reflect mostly free tropospheric air. Previous model calculations suggest very little contrast between daytime and nighttime HCHO columns at Maïdo, due to the long photochemical lifetime of this compound during the night (Stavrakou et al., 2015).

Formaldehyde remains mostly stable between the different sensitivity runs, with similar statistics (Table S4). Being mainly produced from the oxidation of other VOCs, including methane, both the simulated and observed formaldehyde concentrations are highest in January (austral summer), due to higher solar radiation fluxes and biogenic VOC emissions at that time.

### 3.3.2 Methanol

The simulated methanol from the standard run (R0) generally matches the observations very well, except for a slight average positive bias, very similar in January and July (0.2 ppb, see Table S4). This good agreement and the much higher bias (~0.6 ppb) of the S3 run which adopted unadjusted biogenic emission factors (800 µg m$^{-2}$ h$^{-1}$ for all PFTs) appear to validate the halving of the emission factor for the dominant PFTs in run R0 (see Sect. 2.3.2). The lower biogenic emissions are in line with the MEGAN recommendation (Stavrakou et al., 2011). The boundary condition adjustment is also validated, since the

simulation adopting unadjusted values (S2) also displayed higher bias (0.3 ppb) and root mean square error (RMSE) than the R0 simulation. The budget of methanol over the island is largely dominated by the biogenic source, since anthropogenic emissions represent only ca. 1% of the surface emissions of methanol in the model (Fig. 4), and the photochemical production of methanol from the reaction of methylperoxy radicals (CH$_3$O$_2$) with itself and with other peroxy radicals (Jacob et al., 2005) is only a minor source (Müller et al., 2016; Bates et al., 2021). Note that the formation of methanol from the

reaction of CH$_3$O$_2$ with OH (Archibald et al., 2009), which was only recently shown to be a significant source (Bossolasco et al., 2014; Müller et al., 2016; Bates et al., 2021; Caravan et al., 2018), is not included in the model.

Although the R0 simulation performs very well on most days, large positive biases are found during specific periods (e.g. 29-30 January and 2-9 July). Those discrepancies are likely not related to the emissions, since other compounds (e.g. Iox and acetaldehyde) are similarly overestimated during the same period, which suggests an issue with the representation of

meteorology and possibly tracer transport. Cloudiness (in January) and relative humidity (in both months) are overestimated at Maïdo during those periods of low model performance (Fig. 8).

### 3.3.3 Isoprene and monoterpenes

Isoprene is generally well reproduced in WRF-Chem during both months, giving credence to the standard emission factor distribution used in the model. The seasonal variation is correctly predicted, with a factor of 3 higher concentrations in

January compared to July in both the model and the observations. This difference is primarily due to the higher temperatures and radiation fluxes during January (Table S2). Model underpredictions of isoprene are found on days with very low simulated solar radiation due to excessive cloudiness, in particular on Jan. 1-3, Jan. 28-30 and July 27, 29 and 30 (Figs. 12





and 13). Conversely, on days with high observed cloudiness and overestimated modelled radiation fluxes, e.g. on July 7-8, the simulated isoprene is too high. Those patterns reflect the strong control of solar radiation on isoprene emissions in

MEGAN. Duflot et al. (2019) and Verreyken et al. (2021) reported that westerlies are generally associated with higher isoprene abundances at Maïdo, compared to easterlies, presumably because of the closer proximity of vegetation west of Maïdo. However, easterlies were frequently associated with low solar radiation fluxes due to cloudiness, especially in January (1-3 and 27-30 Jan., and to a lesser extent on 9 and 12 July), such that the influence of wind direction on isoprene cannot be established for the period considered here. There is essentially no variability in isoprene concentration between the

different sensitivity runs, except for simulation S4, which predicts slightly higher concentrations around midday (+5% at noon), due to the much weaker OH-recycling in isoprene oxidation in the original MOZART mechanism, compared to the modified mechanism used in R0 (see Sect. 2.2.3). Whereas the average midday isoprene concentration is about 30% too low in the model, possibly due to a small underestimation of the isoprene emission factor, the concentrations in early morning (5-6 UT i.e. 9-10 LT) and late afternoon (17-18 LT) are too high (Fig. 14). This feature, also found for other biogenic

compounds such as methanol, Iox (MVK+MACR+ISOPOOH) and monoterpenes, might be due to insufficient boundary layer mixing.

The model performance for the isoprene oxidation products Iox (MVK+MACR+ISOPOOH) is similar as for isoprene. On many days for which isoprene is too low or too high, so are its oxidation products. There are exceptions to that pattern, such as the first and last days of January, when isoprene is strongly underestimated due to excessive cloud cover, whereas Iox is

not. This difference might be related to wind transport errors in the model, as suggested by the poor model performance for wind direction on those days (see above), or it might be due to the longer lifetime of Iox compared to isoprene (Verreyken et al., 2021) implying that Iox is influenced by more distant sources.

Examination of the average diurnal cycle of Iox concentrations (Fig. 14) shows that, while the PTR-MS data show an almost complete disappearance of Iox during the night, WRF-Chem calculates low, but non-negligible concentrations, especially in

winter. This discrepancy is even larger in the sensitivity simulation S3. While isoprene remains relatively unchanged between the sensitivity runs, Iox shows important changes in the average diurnal shape between R0 and the S3 run (unadjusted biogenic emissions) and S4 run (unadjusted chemistry) . As seen in Table 2, the high-NOx yield of the sum MVK+MACR is much higher (+50%) in the updated chemical mechanism of R0 than in the MOZART-4 mechanism used in S4, which explains the higher Iox concentrations of R0 (by about 10%). Conversely, the simulation S3 predicts significantly

higher Iox levels than R0, by ~30% near midday in July and up to a factor of 5 during the night. The improved diurnal cycle of R0 is due to the strong decrease of the monoterpenes emission factor and their increased light-dependence factor in the R0 simulation, leading to reduced emissions, especially during the night. Indeed, the MOZART-4 mechanism includes a source of MVK and MACR originating from the reaction of lumped monoterpenes with ozone, OH and NO$_3$:

$$C_{10}H_{16} + O_3 \rightarrow 0.7\ OH + MVK + MACR + HO_2 \tag{r8}$$

$$C_{10}H_{16} + OH \rightarrow TERPO2 \tag{r9}$$

$$C_{10}H_{16} + NO_3 \rightarrow TERPO2 \tag{r10}$$





TERPO2 + NO → 0.1 CH$_3$COCH$_3$ + HO$_2$ + MVK + MACR + NO$_2$ (r11)

Since monoterpenes are not a source of MVK or MACR (see e.g., https://mcm.york.ac.uk/MCM/), this artificial source of Iox causes a model overestimation of Iox in both simulations, but especially in simulation S4, and even more during the
night and in July, when the lower radiation fluxes cause a stronger decrease of isoprene emissions, compared to monoterpenes. Both the RMSE and mean bias of R0 are strongly reduced relative to run S4, and an even better agreement would likely be achieved without the artificial source of MVK and MACR from monoterpenes in the model.

The observed ratio of Iox to isoprene concentrations (0.74, based on concentrations averaged between 12 and 16 LT) is reproduced fairly well by the R0 run in January (0.82), whereas a higher ratio (1.05) is derived from run S3, due to its higher
monoterpenes emissions and therefore higher Iox production from their oxidation. In July, the Iox to isoprene ratios are more dissimilar (0.62 based on observations, 1.0 and 1.54 for the R0 and S3 runs), due to the strong reduction of isoprene emissions and higher share of monoterpenes to the the Iox budget in winter, compared to summer.

The much improved agreement for monoterpenes after the biogenic emission adjustments (Table S4 and Fig. 14) appears to validate the lower emissions, especially during the night. A precise assessment is difficult given the large difference between
the PTR-MS concentrations derived from the two signals (m/z 137 and 81, corresponding with protonated monoterpenes and for their main fragments, respectively). These differences might result from e.g. temporal variations in monoterpene distribution or from contributions of other compounds to the m/z signal. The LDF correction improved the temporal correlation between model and observation through the strong reduction of nighttime concentrations, although, as for isoprene, the model simulates early morning and late afternoon concentration peaks that are not seen in the observations. On
average, the modelled values in R0 fall between the two measured signals, i.e. they overestimate (by 8-17 pptv) the monoterpenes detected at m/z 137 and underestimate (by 2-11 pptv) the determination based on the m/z 81 signal.

### 3.3.4 Methyl ethyl ketone and methylglyoxal (m/z 73)

Although both methyl ethyl ketone (MEK) and methylglyoxal (MGLY) contribute to the 73 m/z signal, MEK is dominant according to the model results. As seen on Fig. S3, the modelled MGLY concentrations are highest near local noon, and
account for at most 25% (15%) of the total signal in January (July) in simulation R0, taking into account that the PTR-MS is less sensitive to MGLY (by a factor of 0.7) compared to MEK (Sect. 2.4.2). The observed 73 m/z signal displays a pronounced diurnal cycle (Fig. S34), with daytime concentrations about 3-5 times higher than during nighttime. Although the model simulation without any direct MEK emission (run S5) reproduces this large night/day difference in January, this run strongly underestimates the observations, and it underestimates the amplitude of the diurnal cycle in July (Fig. 14). The
photochemical production of MEK in the model originates exclusively from the oxidation of the surrogate compound BIGALK by OH. The emissions of BIGALK includes the sources of all C$_{\geq 4}$ alkanes, but its oxidation mechanism is that of n-C$_4$H$_{10}$, a well-known major precursor of MEK (Yáñez-Serrano et al., 2016; Jenkin et al., 1997; Sommariva et al., 2011). Among the higher alkanes, only n-butane and 3-methyl propane are significant MEK precursors (Sommariva et al., 2011; https://mcm.york.ac.uk/MCM/). Therefore, since n-butane and 3-methyl propane emissions make up only a fraction of total





BIGALK emissions, of the order of 34% (Stavrakou et al., 2015), the MEK photochemical production in the model is likely
        overestimated. MGLY has no direct source in the model, but it is photochemically produced, mainly from the oxidation of
        isoprene and other BVOCs (Mitsuishi et al., 2018). Given its short lifetime (~1 hour), primarily due to photolysis (Fu et al.,
        2007), MGLY displays a pronounced diurnal cycle with a noon maximum. The molar yield of MGLY in isoprene oxidation
        is of the order of 0.25 (Fu et al., 2007), and it is unlikely to be much underestimated. Therefore, the high MEK/MGLY

mixing ratios observed during the day, ca. 0.07 and 0.05 ppb in January and July, respectively, are best explained by the
        presence of a substantial biogenic source of MEK. This source has been taken equal to 3% of the biogenic isoprene
        emissions in run R0 (see Sect. 2.3.2). The diurnal shape of modelled MEK (run R0) in this run is similar to the observations,
        except that it underestimates the observed diurnal amplitude in July (Fig. S3). Furthermore, in January, the modelled
        concentrations rise too early in the morning and decrease too rapidly in the afternoon, suggesting that the biogenic emissions

of MEK are delayed compared to those of isoprene. This delay is of the order of 2 hours and is qualitatively in line with the
        proposal that MEK is released through the uptake of isoprene oxidation products (MVK and the 1,2-ISOPOOH isomer) by
        vegetation and their subsequent conversion and re-emission as MEK and other compounds (Cappellin et al., 2019; Tani et
        al., 2020; Canaval et al., 2020). The PTR-MS signal for Iox (the sum MVK+MACR+ISOPOOH) is slightly delayed (by 1-2
        h) compared to isoprene, as shown on Fig. S4, and the delay for MVK is expected to be longer than for Iox, due to the lower

rate of the reaction of MVK with OH, compared to MACR and ISOPOOH (e.g. Müller et al., 2019). The adopted ratio
        between MEK and isoprene biogenic emissions (3% on a mass basis) is larger than the value of 1.5% derived by Canaval et
        al. (2020) based on enclosure measurements on gray poplar trees and field eddy-covariance measurements at two forested
        sites. This discrepancy could be partly due to uncertainties and natural variability of MVK/ISOPOOH deposition velocities
        and conversion rates to MEK in plant leaves. In addition, the good model agreement of the R0 run could possibly be

achieved with a lower MEK/isoprene emission ratio, e.g. if the contribution of MGLY to the PTR-MS signal would be
        higher. In addition, the transport of chemical compounds from source regions (e.g. cities) to Maïdo is imperfectly
        represented in the model due to its relatively coarse resolution (2.5km), which may cause errors in the diurnal cycle of
        advected species, in particular for MEK which has a significant anthropogenic component (Fig. 4) (Bon et al., 2011; Brito et
        al., 2015). Finally, note that the model ignores the potentially significant oceanic source of MEK, which is however very

uncertain (Brewer et al., 2020). The S4 run leads to higher MEK/MGLY mixing ratios than R0 around midday (Fig. 14), due
        to differences in the isoprene chemical mechanism (Sect. 2.2.3); the original MOZART-4 mechanism (used in S4) has a
        higher yield of $C_5$-hydroxyaldehydes (HYDRALD), which generate slightly more MGLY than other isoprene oxidation
        products such as MVK and MACR (Emmons et al., 2010).

        In July, the model overestimates both nighttime and daytime observations in run S2, with unadjusted lateral boundary

conditions. This overestimation motivated the adjustment (multiplication by 0.4) of initial and boundary conditions for MEK
        (Sect. 2.2.4). This adjustment has much less effect in January, due to its shorter photochemical lifetime in summer, compared
        to winter.



### 3.3.5 Acetaldehyde

Regarding acetaldehyde, WRF-Chem largely underestimates the PTR-MS concentrations in January, whereas the model is
essentially unbiased on average in July (-0.009 ppb), but correlates very poorly with the observations ($r$=0.39, Table S4).
The temporal correlation improves markedly (to ca. 0.75 in January and July) when the direct emissions of acetaldehyde and
other VOCs are turned off (run S5), even though the mean bias increases. This finding suggests that direct emissions of
acetaldehyde are overestimated, while photochemical production might be too low. Acetaldehyde emissions in the model are
largely due to the residential sector (Fig. 4). Aldehydes are the main NMVOC class released by the residential sector, and
these emissions are mainly due to biomass burning combustion (Huang et al., 2017). The model uses the combined
emissions of all higher ($C_{\geq 2}$) aldehydes as anthropogenic emissions of $CH_3CHO$. However, acetaldehyde accounts for only a
small fraction of total aldehyde emission from boilers (7-17%) (Macor and Pavanello, 2009) and from the heating of cooking
oils (at most ~1%) (Katragadda et al., 2010; Takhar et al., 2023). The anthropogenic emissions of $CH_3CHO$ used in the
model are therefore strongly overestimated, possibly by one order of magnitude or more. Without this lumping, we expect
that the model would underestimate the PTR-MS data by almost one order of magnitude, suggesting that other sources of
acetaldehyde are likely strongly underestimated. This would be in line with previous assessments based on in situ data at
remote marine locations (e.g., Wang et al., 2019; Travis et al., 2020). For example, at Cape Verde in the Atlantic Ocean, the
CAM-Chem model was found to underestimate in situ concentration measurements by a factor 30 (Read et al., 2012), and
the discrepancy was attributed to both oceanic acetaldehyde emissions and photochemical production from unknown,
relatively long-lived precursors. Direct oceanic emissions might contribute significantly to the signal at Maïdo (Millet et al.,
2010; Read et al., 2012), given the relative proximity of the sea (~20km) and the relatively long lifetime of $CH_3CHO$ (2-9
hours, with [OH] = $(2-10)\times10^6$ molec. cm$^{-2}$, see Fig. 9). Photochemical production of acetaldehyde in the model proceeds
mainly from the oxidation of alkanes, alkenes and ethanol (Millet et al., 2010). The total production of $CH_3CHO$ calculated
using the model (R0 simulation) over Réunion Island averaged over January and July amounts to 19, 0.92, 14, 13, 4.2, 3.86
and 18 tons per month for ethane, propane, propene, higher alkenes (BIGENE), higher alkanes (BIGALK), MEK and
ethanol, respectively. Unfortunately, only MEK and ethane were measured at Maïdo, and large model underestimations
cannot be excluded for any of the other precursors. For example, the average mixing ratios of propane and BIGALK
calculated by the model at Maïdo are only ca. 20 and 30 ppt, respectively, whereas long-term in situ measurements at other
southern tropical marine sites (Seychelles and Ascension) are considerably higher, of the order of 100 ppt in January and 50
ppt in July for propane, and about 150-250 ppt in January and 100-200 ppt in July for the total concentration of measured
butanes and pentanes (n-$C_4H_{10}$, i-$C_4H_{10}$, n-$C_5H_{12}$, i-$C_5H_{12}$) (Helmig et al., 2021). Biogenic emissions of acetaldehyde and
ethanol are included in the model but are very uncertain (Millet et al., 2010) and might be underestimated. As suggested by
Travis et al. (2020), the oceanic emissions of various acetaldehyde precursors (primarily ethanol, alkanes, alkenes) might
partially explain the negative model biases against aircraft in situ data across the globe and particularly at remote locations
and in the free troposphere. These emissions are currently ignored in the model simulations.





### 3.3.6 Acetone

Due to its relatively long lifetime (several weeks) (Jacob et al., 2002; Fisher et al., 2012), acetone is sensitive to the lateral boundary conditions, especially in July when solar radiation and OH levels (Fig. 9) are at their minimum. The large positive model bias of the S2 run in July is reduced by a factor of 10 after adjustment of boundary conditions (R0 run). Acetone is relatively insensitive to this change in January, likely due to its shorter lifetime and larger biogenic emissions during summer. Acetone is sensitive to a removal of its direct emissions (S5), which have a small anthropogenic component (~3 tons/month) and a dominant biogenic contribution (55 tons/month, Fig. 4). The secondary source of acetone from photochemical production is comparatively weak over the island (17 tons/month on average for the two months). The principal precursors are BIGENE, BIGALK, propane and monoterpenes, with productions amounting to 8.5, 3.4, 3.5 and 1.7 tons, respectively. As in the case of acetaldehyde, the contribution of propane and BIGALK oxidation is likely strongly underestimated. In agreement with previous studies, (e.g. Wang et al., 2020), the direct biogenic source of acetone is dominant over the secondary production due to monoterpenes and anthropogenic VOCs. This source is poorly constrained and therefore very uncertain, the acetone emission factor being constant for all PFTs in MEGAN, except for grasses and crops (Guenther et al., 2012). The ocean/atmosphere exchange of acetone plays a substantial role over the remote troposphere (Wang et al., 2020) but is neglected in our study. The diurnal cycle of acetone mixing ratios is underestimated by the model during both months (Fig. 14). Since photochemical production plays only a minor role for this compound, this underestimation might be due to either a misrepresentation of transport patterns (Fig. 8) or an underestimation of biogenic emissions during daytime. The light-dependent fraction (LDF) for biogenic acetone is only 0.25 in the model, implying little diurnal variability of the emissions. Comparison of the model performance for acetone (Fig. 12) with the meteorological evaluation (Fig. 8) does not suggest a clear link between the modelled diurnal cycle and meteorological parameters, in particular wind direction.

### 3.3.7 Acetic acid, peracetic acid and glycolaldehyde (m/z 61)

The m/z 61 signal exhibits a pronounced diurnal shape in January and July, with a clear daytime maximum, and nighttime values approximately 5 times lower than the midday peak (Fig. 14). Although this signal is mainly attributed to acetic acid, there are interferences, mainly from glycolaldehyde (GLYALD) and peracetic acid (PAA). Here, this measurement is compared against the sum of acetic acid, GLYALD and PAA from WRF-Chem. The contributions of the three species to the combined modelled concentration are displayed on Fig. 15. The model strongly underestimates the magnitude of the signal, by a factor of 3-4, but it replicates the observed diurnal and seasonal variation. All three species have their highest concentrations during daytime in January, when photochemical activity and biogenic emissions are at their highest. Acetic acid has only a weak biogenic source (5 tons/month on average over the island, Fig. 4), in agreement with previous studies (e.g. Paulot et al., 2011), while all three compounds have significant photochemical production terms. GLYALD is produced at a high yield (~0.3 at high NOx) by the oxidation isoprene by OH and has a short chemical lifetime (a few hours during the





day), which explains that its nighttime concentrations are very low (Fig. 15). Given the large isoprene emissions, GLYALD is the main contribution to m/z 61 during the day according to the model. The oxidation of monoterpenes also leads to

835 GLYALD formation in the MOZART-4 mechanism, which explains the higher GLYALD concentrations of the S3 run, which uses unadjusted (e.g. 5-times higher) emissions of monoterpenes.

The contribution of the two acids shows less diurnal variation than GLYALD, particularly in July. Photochemical production of these two compounds over the island amounts to 19.5 and 62.4 tons/month for acetic acid and PAA, respectively, on average for the two months. It is mainly due to the reaction of the acetylperoxy radical ($CH_3CO_3$) with $HO_2$, represented in

the model as

$$CH_3CO_3 + HO_2 \rightarrow 0.75\ PAA + 0.25\ CH_3COOH + 0.25\ O_3 \tag{r12}$$

This reaction alone accounts for 79% and 94% of the total production of acetic and peracetic acid over the island, respectively. The model ignores the radical-forming channel,

$$CH_3CO_3 + HO_2 \rightarrow CH_3CO_2 + OH + O_2 \tag{r13}$$

(followed by $CH_3CO_2$ ($+O_2$) $\rightarrow CH_3O_2 + CO_2$), which was shown to account for about half of the total reaction (Jenkin et al., 2007). The total rate constant of the reaction, $1.4 \times 10^{-11}$ molec.$^{-1}$ cm$^3$ s$^{-1}$ at 298 K in MOZART-4, is also likely underestimated by a factor of ca. 1.5 compared to more recent estimates (Gross et al., 2014). These approximations might contribute to a moderate overestimation of acetic and peracetic acid production from the reaction above, although the probable underestimation of alkanes and other VOCs noted above likely implies an underestimation of the production rate of

$CH_3CO_3$ radicals. In addition, many reactions of $CH_3CO_3$ radicals with other organic peroxy radicals are neglected in the mechanism, although they might contribute significantly to acetic acid production (Khan et al., 2018). Furthermore, the rate constant of the reaction of peracetic acid with OH was recently shown in a combined experimental and theoretical study (Berasategui et al., 2020) to be much lower (factor of ~30) than previously determined, implying a much longer lifetime in the troposphere.

The model underestimation of the m/z 61 signal is consistent with (1) the suggestion of underestimated or unknown sources of acetic acid based on model comparison with in situ measurements from aircraft and at surface stations (Paulot et al., 2011; Khan et al., 2018), and (2) the observations of high PAA concentrations over remote oceans during the Atmospheric Tomography Mission (ATom) (Wang et al., 2019). Given that acetaldehyde is a known major precursor of PAA, box model calculations by Wang et al. (2019) showed that these high levels of PAA are fully consistent with the (simultaneously

observed) acetaldehyde measurements. As noted above, there is abundant evidence of a large source of acetaldehyde over the oceans.





**Figure 12. Time series of observed and modelled concentrations of PTR-MS measured species in January. All concentrations are shown in ppb. The model results are shown for simulations R0, S2, S3, S4 and S5 (see Table 4). Both monoterpene measurements are depicted, differentiated by solid (m/z 81) and dashed black lines (m/z 137).**





**Figure 13. Same as Fig. 12, for July 2019.**





Figure 14. Average diurnal cycle of organic compounds measured at Maïdo observatory, and corresponding modeled concentrations from the 30-day runs defined in Table 4 (except S1). The concentrations are shown in ppb. Both monoterpene measurements are depicted on the same figure, differentiated by solid (m/z 81) and dashed black lines (m/z 137).



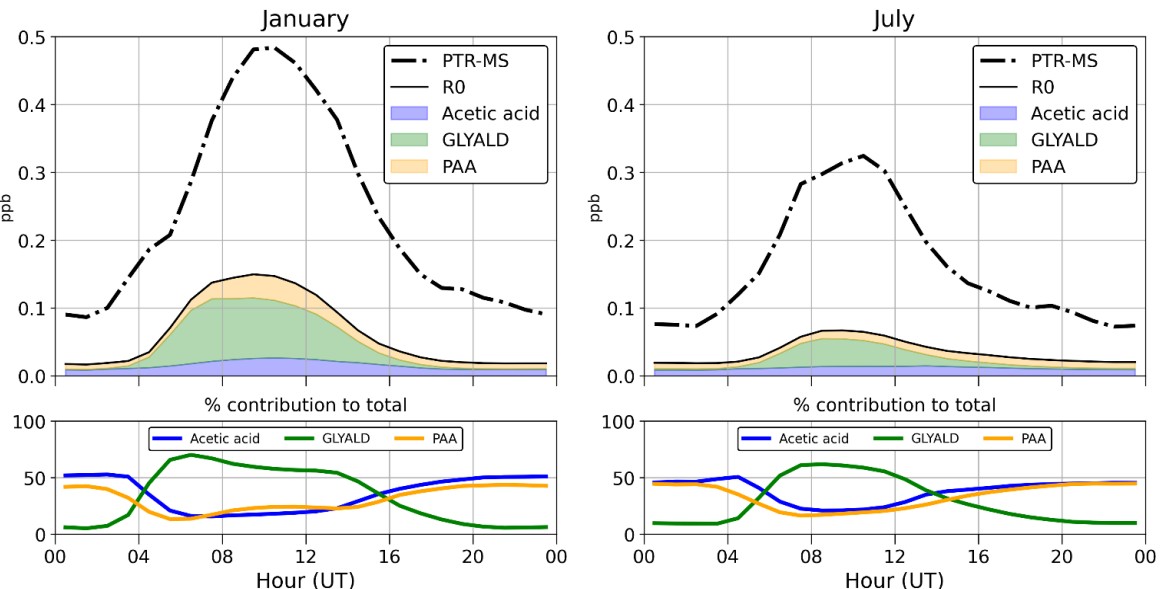

**Figure 15. Averaged diurnal cycle of the m/z 61 signal in January (left) and July (right), and comparison with the modelled concentrations of acetic acid, GLYALD and PAA, which are all assumed to contribute to the observed signal. The PTR-MS sensitivity is assumed identical for the three compounds.**

### 3.4 Evaluation against FTIR columns

The model is evaluated against FTIR data in Fig. 16. The model values are sampled at the same times as the measurements and smoothed using the averaging kernels and a priori profile of the FTIR retrieval. The results of two simulations are shown: R0, the standard run with all model updates, and S2, identical to R0 but without the adjustments to the initial and boundary conditions (ICBC). The results of the other simulations differ only marginally from those of the R0 run, and are therefore not shown. ICBC adjustments were made only for $CH_3OH$, PAN and $C_2H_6$; for the other compounds, the discrepancies are generally small.

For HCHO, a model overestimation is noted during both months, amounting to 8 and 2 $\times 10^{14}$ molec. $cm^{-2}$ in January and July, respectively. In January, this bias exceeds the reported systematic uncertainty of the FTIR column ($\sim 2.5 \times 10^{14}$ molec. $cm^{-2}$) (Vigouroux et al., 2018) and is not likely due to ICBC errors for HCHO, given the short lifetime of this compound (a few hours). The monthly averaged retrieved columns (3.2 and $1.61 \times 10^{15}$ molec. $cm^{-2}$ in January and July, respectively) are similar to the reported FTIR columns at Maïdo for the same months in previous years (Vigouroux et al., 2018; 2020). The weak sensitivity of the modelled columns to changes in local VOC emissions (e.g. from run S5) suggests that the cause of the model overestimation is due either to HCHO precursors that are not well represented in the model, or to background HCHO, i.e. primarily the contribution of methane oxidation, above $\sim 2$ km altitude (the altitude of Maïdo). The addition of lightning NO emissions (run S6) aggravates the bias, as it increases the HCHO column (+24% in January, see Fig. S5). The HCHO enhancement due to lightning NO is mostly located in the mid-troposphere, between 4 and 11 km (Fig. S6) and is primarily caused by the increase in OH consecutive to the added $NO_x$ emissions in this altitude range, as the OH radical



promotes methane oxidation and its associated HCHO production. At lower altitudes, OH levels are less impacted, whereas at higher levels, HCHO is more controlled by deep convection of low-altitude pollutants to higher altitudes (Bozem et al., 2017; Fried et al., 2008).



**Figure 16. Time series of FTIR-measured and modelled HCHO, CH₃OH, C₂H₆, CO, PAN and tropospheric O₃ columns in January**
**(top rows) and July (bottom rows) 2019. The symbols denote the individual FTIR measurements and corresponding WRF-Chem results, while the solid lines (red for R0, cyan for S2) show the 24-hr averages.**

Given the longer atmospheric lifetimes of the other compounds shown on Fig. 16, the model performance for these species is very dependent on their boundary and initial conditions, obtained from either CAM-Chem (for CH₃OH) or CAMS (for the other compounds). Without any adjustment, the model agreement with observed CO and tropospheric O₃ columns is

reasonable, with biases generally well below 10% for CO and 20% for O₃. The simple ICBC adjustments, presented in Sect.





2.2.4, succeed generally very well in reducing the gap between model and observations for the other compounds. Since the ICBC adjustments consist of a single scaling factor for each species, the short-term variability of the observations remains poorly reproduced by the model after adjustment, e.g. for $CH_3OH$. Nevertheless, the model correlates well with the daily-averaged data for $O_3$ (Pearson's correlation coefficient $r$=0.6 and 0.7 in January and July), and to a lesser extent also for

$C_2H_6$ ($r$=0.7 and 0.3).

The monthly averaged FTIR PAN columns, ca. $1\times10^{15}$ molec. cm-2, are lower, but of the same order of magnitude as those typically observed above the Jungfraujoch station in the Swiss Alps during summer (up to $4\times10^{15}$ molec. cm$^{-2}$) (Mahieu et al., 2021). They are also significantly lower than those observed by the Infrared Atmospheric Sounder Interferometer (IASI), typically $>2\times10^{15}$ molec. cm$^{-2}$ above oceanic areas around Réunion Island (Franco et al., 2018), although this difference is

likely partly due to the high altitude of the station. The spatial distribution displayed by IASI suggests a substantial influence of long-range transport of African emissions on PAN levels in the region. Although PAN is short-lived in the lower troposphere due to fast thermal decomposition, it is much longer-lived in the cold upper troposphere, where it can be transported over long distances. Due to this long lifetime, a large fraction of the column lies at those high altitudes, which explains why the modelled PAN column at Maïdo responds strongly to the ICBC change between simulations S1 and R0.

Note that this does not imply a strong influence of ICBC on PAN and NOx close to the surface, where the PAN lifetime is short. The modelled columns of the S1 run overestimate the FTIR data by about a factor 2, providing the justification for the adjusted ICBC for PAN in simulation R0. However, both FTIR data and the R0 run underestimate IASI columns by a factor of 2 or more. The acknowledged uncertainties of the PAN retrievals by FTIR and IASI suggest that the ICBC adjustment for PAN should be considered as very uncertain.

## 3.5 TROPOMI

The distribution of $NO_2$ and HCHO columns obtained from the oversampling of TROPOMI retrievals over a period of more than four years is displayed on Fig. 17. Despite the much lower $NO_2$ columns retrieved over and around the island ($<1.5\times10^{15}$ molec. cm$^{-2}$) in comparison with e.g. Europe (typically $(2-10)\times10^{15}$ molec. cm$^{-2}$) (Poraicu et al., 2023; Lange et al., 2023), sharp gradients are observed, reflecting the strong heterogeneity of $NO_x$ emissions. A very well-localized

maximum ($1.5\times10^{15}$ molec. cm$^{-2}$) is found at the precise location of the Le Port power plants (Fig. 17), whereas the other thermal power plants, located near Sainte-Suzanne (Bois Rouge) on the northern coast, and near Saint-Pierre (Le Gol) on the southern coast, are not clearly detected. The emissions from the Le Port power plants (and to a lesser extent from the cities of Le Port, Saint-Denis and Saint-Paul) generate a broad hot spot of $NO_2$ columns exceeding $1\times10^{15}$ molec. cm$^{-2}$, which extends over land as well as over sea over distances of the order of 5-10 km. This distribution confirms the dominance of the Le Port

power plants emissions over those from the other point sources, and justifies the redistribution of industrial emissions based on the EDGAR 6.1 dataset described in Sect. 2.3.1. The $NO_2$ column values around Maïdo are significantly lower, of the order of $0.5\times10^{15}$ molec. cm$^{-2}$. Even lower columns are observed in the southeastern part of the island, around the volcano (Piton de la Fournaise).





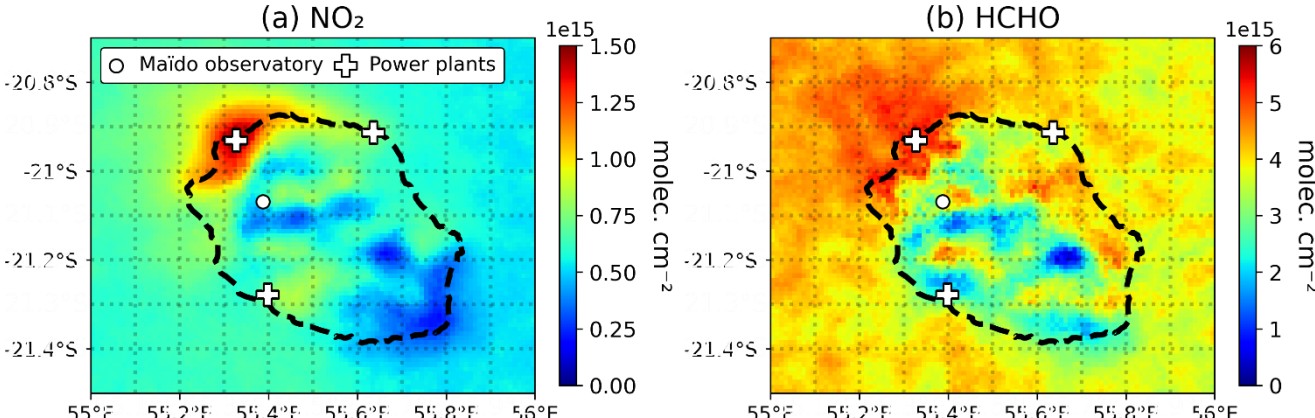


**Figure 17. Oversampled NO₂ and HCHO columns from TROPOMI, covering 2018-2022. The resolution is 0.01°×0.01°.**

The oversampled distribution of TROPOMI HCHO is noisier compared to NO$_2$, but also shows well-defined features and gradients. The largest columns (~5×10$^{15}$ molec. cm$^{-2}$) are found in a large area along the northwestern coast and in particular near the Le Port power plants. Since the power plants are not a large VOC source, and since the HCHO distribution shows little correlation with the anthropogenic VOC emission distribution (Fig. 6), these high columns clearly do not reflect the distribution of VOC emissions. Instead, they are likely due to the influence of anthropogenic NOx leading to higher OH levels and therefore to enhanced HCHO production rates from the oxidation of long-lived hydrocarbons, especially methane. Methane oxidation is by far the largest source of HCHO at the global scale (Stavrakou et al., 2009), and to a large extent also above Réunion Island (see further below), even though non-methane hydrocarbons are the dominant source over most continental areas. This dominance of the background is the main reason for the many common features shared by the HCHO and NO$_2$ distributions from Fig. 17, such as the high values to the west and northwest of the island, and the lower columns to the east and southeast of the island and over several specific areas in the island interior. Besides this likely dominant contribution of the methane background to HCHO, the TROPOMI distribution also shows evidence of elevated columns due to isoprene emissions, in particular along the eastern coast (see Fig. 5).

The model is evaluated against NO$_2$ columns from TROPOMI in January and July 2019 in Fig. 18. The domain shown on Fig. 18 encompasses both Réunion and Mauritius Island, located to the Northeast of Réunion, at a distance of ca. 200 km. The results of two simulations are shown: the reference run without lightning (R0) and the simulation with lightning emissions (S6). The modelled columns of run S1 (with higher NOx emissions from the Le Port power plants) are given in the Supplement (Fig. S7). The TROPOMI averaging kernels were applied for both compounds, and the model averages were calculated from values sampled at the same times as the TROPOMI monthly averages. Both modelled and retrieved columns were regridded to 0.1° resolution.

Both the model and TROPOMI data display higher NO$_2$ columns on Mauritius Island, compared to Réunion. Based on EDGARv6.1 emission data, the main hotspot on Mauritius is due to the energy sector, with several fossil fuel power plants located in Port-Louis, the capital city. The model comparison with TROPOMI strongly suggests that the EDGAR NOx



emissions are overestimated. Those emissions generate a large plume extending towards the Southwest (in January) or the
West (in July), but its impact on Réunion appears to be limited, based on the distributions shown on Fig. 18.

Over Réunion, both R0 and S6 model simulations reproduce the main spatial patterns observed in the measurements,
namely, the maximum at Le Port and over adjacent regions, primarily the eastern and (to a lesser extent) the southeastern
coast in January, and the northern coast, in July. A secondary $NO_2$ maximum is also seen along the southern coast near
Saint-Pierre and especially the thermal power plant of Le Gol. The model also reproduces the minimum seen in the eastern
part of the island, in January, and near the southeastern extremity of the island, in January. Overall, both R0 and S6
overestimate the observations in the winter, but there is an underestimation for R0 during summer (-11% and +64% for the
island-averaged column of R0 and S6 in January, respectively and +25% for both in July). The discrepancy is however
locally much higher, e.g. near Maïdo in July (factor of ~1.8 at the two pixels nearest to Maïdo), likely due to the export of
pollution from the Le Port area. Caution is warranted when comparing model and satellite over remote locations, as the
satellite columns are very noisy, despite their spatial (0.1°) and monthly averaging, especially over the mountainous island
interior. The TROPOMI $NO_2$ uncertainties are of the order of $5\times10^{14}$ molec. cm$^{-2}$ (see Fig. S8), i.e. they are of the same
order as the retrieved columns over most of the island except the strongest anthropogenic hotspots. The model correlates
better with the data in July ($r = 0.72$ for R0 over the inner domain, see Fig. 18) than in January ($r = 0.43$), likely due to the
higher values in July. The model reproduces the observed seasonal variation of $NO_2$ columns, with higher values found in
July (winter) compared to January (summer), in response to changes in OH radical concentrations (Fig. 9). The R0 run
provides a significantly better match with TROPOMI than the S1 run, which strongly overestimates the columns along the
western and southwestern coast (Fig. S7), by a factor of 2, as well as over the sea, west and southwest of the island. Those
overestimations essentially disappear in the R0 run, which assumes a 5-fold reduction of the NOx emissions due to the Le
Port power plants.

The causes for the slight $NO_2$ model overestimation of the R0 run in July are unclear. The NOx emissions could be
overestimated, the NOx lifetime could be too long, or the TROPOMI $NO_2$ columns might be too high. Unfortunately, most
TROPOMI $NO_2$ validation studies were conducted at mid-latitudes, in regions with strong pollution sources (Verhoelst et al.,
2021; Poraicu et al., 2023; Lange et al., 2023; van Geffen et al., 2022a). Those studies have generally reported the slope (s)
and intercept (i) of linear regressions of the type C = i + s C', where C denotes the TROPOMI column and C' the co-located
independent measurement. Most validation studies reported positive values of the intercept i and slope values s lower than
unity, suggesting that TROPOMI underestimates high values and overestimates very low values. More studies are needed to
better characterize the potential biases of TROPOMI $NO_2$ in remote areas.



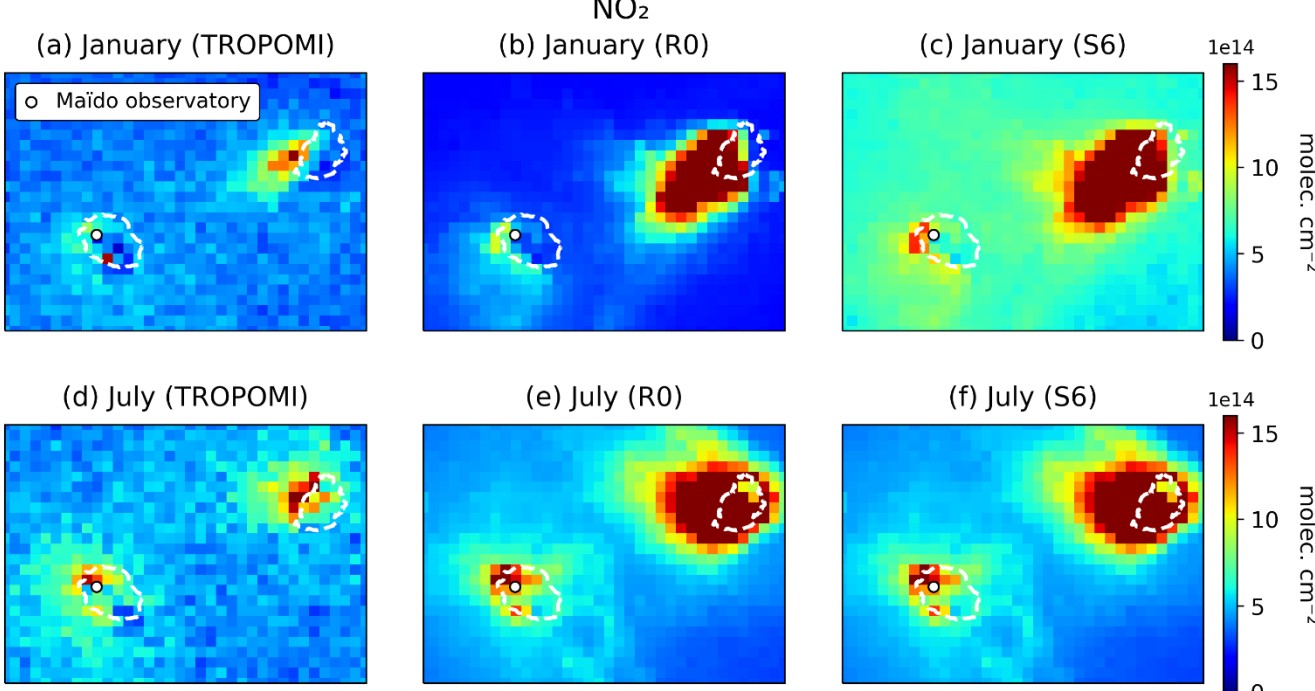

**Figure 18. Monthly-averaged NO₂ columns from TROPOMI and WRF-Chem (simulations R0 and S6) for January (top row) and July (bottom row), regridded to 0.1° resolution. The WRF-Chem averages account for the TROPOMI sampling times and for the averaging kernel corresponding to each overpass. The white line denotes the boundary of the inner model domain (d02).**

Implementation of lightning emissions (run S6) leads to a strong column enhancement in January, while its impact is negligible in July. The S6 run worsens the model overestimation over land, and leads to overestimated columns over ocean. For example, the average column from run S6 over an oceanic area north of Réunion is $6.3 \times 10^{15}$ molec. cm$^{-2}$, a factor of 1.75 above the TROPOMI average of $3.61 \times 10^{15}$ molec. cm$^{-2}$, whereas the R0 run is a factor of 2 too low ($1.83 \times 10^{15}$ molec. cm$^{-2}$). The lightning emissions flip the seasonality of NO₂ columns over ocean, with higher values predicted in January, while both TROPOMI and the R0 run suggests a wintertime maximum. In January, lightning NOx is responsible for increasing the average mixing ratio of NO₂ in the upper troposphere by a factor of 4, compared to the R0 simulation (Fig. S6). Note that the default WRF-Chem settings would lead to much larger lightning emissions since, following Barten et al. (2020), we decreased the number of flashes by a factor of 10, and we also reduced the NO production per flash to 250 moles. Nevertheless, the S6 run substantially overestimates the upper tropospheric (UT) NO₂ mixing ratios obtained from TROPOMI using the cloud slicing technique (Marais et al., 2021; Horner et al., 2024). Indeed, the TROPOMI-based UT mixing ratios (between 180 and 450 hPa) are typically 40-50 pptv above the Indian ocean around Réunion Island during Dec. 2019 - Jan. 2020 (Marais et al., 2021), whereas the model averages over this region are 28 and 120 pptv in the R0 and S6 runs, respectively. Both the model comparison with the tropospheric columns (Fig. 18) and with the UT mixing ratios based on TROPOMI indicate that the lightning source of the S6 run is too high, despite the reduction of flash count and number of



moles per strike. In July, lightning emissions are largely insignificant, with less than 2.5% increase in NOx in the upper troposphere.

Much like for TROPOMI $NO_2$, the model evaluation against TROPOMI HCHO data shows a relatively good agreement with regards to spatial representation (Fig. 19), even though the spatial correlation coefficients are very low (e.g. $r = 0.2$ for the R0 run in January), largely due to the high noise in the data. In January, relatively high columns (of the order of $8 \times 10^{15}$ molec. cm$^{-2}$ in the R0 run, slightly less in the TROPOMI columns) are found to the southwest of the two islands, as well as along the western coast of Réunion. These patterns mirror the $NO_2$ distribution, which also shows enhancements in those

areas (Fig. 18). Nevertheless, HCHO columns from both R0 and S6 show overestimations when compared to the observations. For example, TROPOMI shows values close to $6 \times 10^{15}$ molec. cm$^{-2}$ along the northwestern coast of the island in January, slightly lower than the modelled values close to $8 \times 10^{15}$ molec. cm$^{-2}$. The discrepancy is close to the uncertainty provided with the retrieval product, $2 \times 10^{15}$ molec. cm$^{-2}$ for the systematic component (trueness) and ca. $1 \times 10^{15}$ molec. cm$^{-2}$ for the random part (precision), when accounting for the number of measurements used in the shown averages (see Fig. S9).

This model overestimation is consistent with the model overestimation against FTIR HCHO columns at Maïdo (see Sect. 3.4). As pointed out above, the main source of HCHO in this region is methane oxidation. Based on the model output, this production is estimated at about 3.0 and 1.2 Gg per month over the land area of Réunion for January and July, respectively. The overestimation of HCHO levels in the model, in comparison with TROPOMI and FTIR data, can probably be explained by an overestimation of OH levels in the model simulations. Note that the addition of lightning in S6 increases the OH

levels, and therefore worsens the overestimation of HCHO columns in January. There might be several causes of the HCHO overestimation. NOx might be too high, as suggested by the moderate overestimation of $NO_2$ columns against TROPOMI data, and this could impact OH. In addition, the neglect of halogen chemistry in the model likely also leads to OH overestimation (Sherwen et al., 2016). Finally, it was recently found out that the absorption of UV radiation by water vapor is more significant than previously assumed, and slows down the rate of ozone photolysis and therefore the production and

concentrations of OH radicals in the lower troposphere (Prather and Zhu, 2024). This process is particularly efficient at tropical latitudes where water vapor is most abundant. Note however, that this effect might be partially counterbalanced by the reduction of HCHO photolysis rates, following the decrease of UV radiation levels.

The HCHO production due to methane oxidation might be compared with the contribution of isoprene, a major source of HCHO at the global scale. Taking an average isoprene flux of 0.25 Gg month$^{-1}$ over the island (Fig. 4) and assuming 2.5

HCHO molecules produced for one molecule of isoprene (Stavrakou et al., 2009), the HCHO production from this source is estimated to be 0.28 Gg month$^{-1}$, i.e. almost an order of magnitude lower than the contribution of methane oxidation (see previous paragraph). The secondary source of HCHO from anthropogenic VOC oxidation might be significant, but the lifetimes of these compounds being much longer compared to isoprene, the HCHO production over the island is diluted due to mixing, and partially exported away from the island.





Whereas the model overestimation of HCHO columns with respect to TROPOMI might be partially due to model errors, the TROPOMI columns are very low and noisy, and therefore close to the detection limit of the instrument. Taking the detection limit as three times the precision, most frequently in the range $(0.5\text{-}1.5)\times10^{15}$ molec. cm$^{-2}$ for $0.1°$ pixels over the island (Fig. S9), and taking into account that the TROPOMI precision was found to be underestimated by a factor of about 2 (1.6-2.3) in comparisons against FTIR data from a wide network of stations (Vigouroux et al., 2020), almost all TROPOMI monthly

averages over the island (except around the Le Port hotspot) fall below the detection limit, which rationalize the high noise seen on the TROPOMI HCHO maps (Fig. 19).

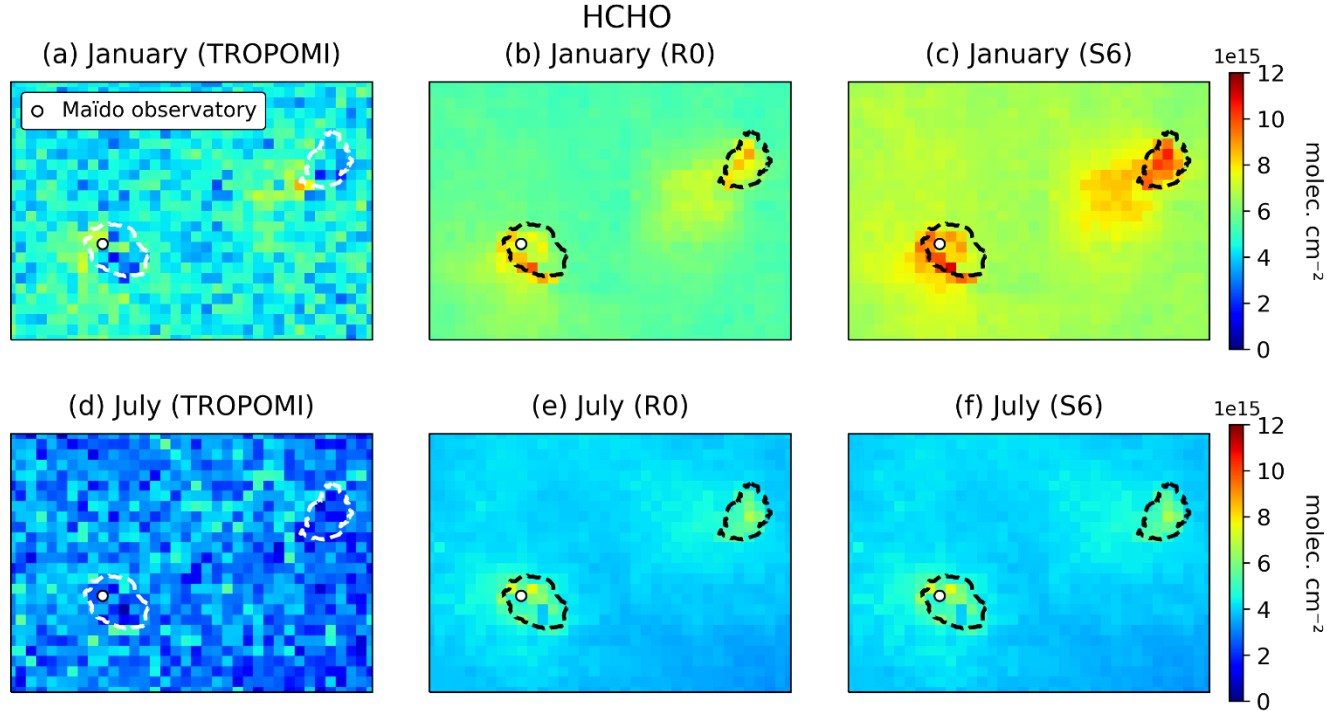

**Figure 19. Monthly-averaged formaldehyde columns from TROPOMI and WRF-Chem (simulations R0 and S6) for January (top row) and July (bottom row), regridded to $0.1°$ resolution. The WRF-Chem averages account for the TROPOMI sampling times**
**and for the averaging kernel corresponding to each overpass. The white dotted line denotes the borders of the inner model domain (d02).**

## 6. Conclusions

The WRF-Chem model has been used to compute the atmospheric composition in a region surrounding Réunion Island, in January and July 2019. The oversampled TROPOMI NO$_2$ distribution demonstrates the predominant influence of
anthropogenic emissions on NOx abundances over the island. In particular, the power plants near Le Port cause by far the strongest NOx hotspot over the island, implying that their emissions largely exceed the emissions of any other point source or major city over Réunion. The model (R0 simulation) is only moderately biased against TROPOMI NO$_2$ columns (-11% in



January, +25% in July) and reproduces the main observed patterns. To reach this agreement, the high-resolution (1 km$^2$)
anthropogenic emission inventory used as input to the model has been adjusted with regard to the repartition of energy sector

emissions among the different power plants (in accordance with EDGAR v6.1), and furthermore, the NOx emissions due to
the Le Port power plants were downscaled by a factor of 5. This reduces the industry/energy sector emission estimates from
Atmo-Réunion and EDGAR v6.1 by factors of 3.8 and 1.9, respectively. The emissions from the other power plants could
not be constrained based on TROPOMI data. The 5-fold reduction above is a crude adjustment, but it is comforted by the
model comparisons with both TROPOMI NO$_2$ and in situ NOx measurements in the direct vicinity of the Le Port power

plants (Fig. 2). The fairly good model agreement against in situ NOx measurements at other air quality stations (Figs. 10-11)
lends confidence in the anthropogenic NOx emission estimates for the other sectors (mostly traffic). In situ O$_3$ concentrations
are overestimated by the model at the air quality stations, by ca. 6 ppb on average. This is likely due to the neglect of
halogen (Cl, Br, I) emissions and chemistry in the model. These compounds were shown to deplete O$_3$ substantially in the
tropical marine troposphere, by about 7 ppb (e.g. Badia et al., 2019).

Over the ocean, the model comparisons with TROPOMI (for January) show the importance of the lightning source of NO.
Accounting for this source (S6 run), or neglecting it (R0) leads to either overestimated or underestimated modelled NO$_2$
columns in comparison to TROPOMI over sea (Fig. 18); the comparison against upper tropospheric NO$_2$ mixing ratios based
on cloud-sliced TROPOMI NO$_2$ data follows the same trend, and confirms the significant impact of lightning in this region
in January. However, in agreement with Barten et al. (2020), the lightning emissions are strongly overestimated by WRF-

Chem with its current parametrization, even when the flash count is reduced by an order of magnitude in the model.
The model performance against the PTR-MS VOC measurements is species-dependent, and the model comparisons
prompted several adjustments to the MEGAN model-calculated emissions. Most importantly, the biogenic emissions of
methanol and monoterpenes were downscaled by factors of about 2 and 5, respectively, and the light-dependent fraction for
monoterpenes was increased to 0.9 (see Sect. 2.3.2), in order to provide a fair match with the measurements. The model

performs quite well for isoprene and for its oxidation products (Iox), but their diurnal shape displays unrealistic peaks in the
early morning and late afternoon (a feature also seen for other compounds), likely due to an underestimation of vertical
mixing in the model. The ratio of Iox to isoprene (ca. 0.8 around noon in January) is fairly well reproduced. This is
reassuring regarding the model performance, since the ratio is strongly dependent on oxidative conditions, i.e. on OH levels.
Despite a good reproduction of the seasonal variation and diurnal shape of the formaldehyde concentrations observed by

PTR-MS, the model underestimates the measurements, by up to 25% during the day and a factor of 2 during the night. This
stands in contrast with the model overestimation of HCHO column measurements by FTIR (also at Maïdo) and TROPOMI,
for reasons unclear. The discrepancy against the column measurements could be due to an overestimation of the background
HCHO production, primarily due to methane oxidation, which would result from overestimated OH levels in the free
troposphere. The underestimation against PTR-MS could possibly indicate direct HCHO emissions; however, the nighttime

measurements at Maïdo are primarily influenced by the free troposphere. More work will be needed to elucidate the factors
influencing formaldehyde concentrations around Maïdo.





The model fails to reproduce both the seasonal cycle of acetaldehyde (higher values observed in summer compared to winter) and its diurnal variation (strong midday peak). The anthropogenic source of $CH_3CHO$ in the model is almost certainly too high, due to the lumping of higher aldehydes into this compound. Lower anthropogenic emissions and a much higher photochemical production would be needed to match the observations. The modelled concentration of $C_{\geq 3}$ alkanes is very likely underestimated, as shown by the model evaluation against measurements in similar environments. Therefore, as proposed in previous studies, a large part of the missing production of acetaldehyde may arise from the oxidation of alkanes, alkenes and alcohols, presumably released by the oceans.

Methyl ethyl ketone (MEK) being the largest contribution to the 73 m/z signal from the PTR-MS, the large observed daytime mixing ratio suggests the presence of a significant biogenic source of MEK, well in line with the proposal that the deposition of isoprene oxidation products (MVK and 1,2-ISOPOOH) on vegetation is followed by their partial conversion and re-emission as MEK. The ratio between MEK and isoprene emission which achieves the best agreement with the data is 3% (mass basis), higher than the value of this ratio derived by Canaval et al. (2020) based on more direct emission measurements, for different environments. The difference with the latter study might be due to natural variability and/or to model uncertainties related e.g. to other contributions to the observed signal.

Clearly, more work will be needed to understand and quantify the budget of key compounds ($O_3$, OH, NOx, VOCs and OVOCs) around Réunion Island and similar environments. From the modelling perspective, efforts should be made to improve the representation of NMVOC speciation, with more explicit chemical mechanisms (e.g. for monoterpenes and aromatics) and more detailed emission inventories (e.g. for higher aldehydes). The advection of chemical compounds should be improved, e.g. with finer resolution modelling to better represent transport in a mountainous environment, and the vertical mixing of the WRF-Chem model (especially during nighttime) should be addressed, e.g. as proposed in Kuhn et al. (2024). The ocean/atmosphere exchange of important VOC should be implemented in the model, as it is a well-recognized source of several OVOCs and their precursors (alkanes, etc.). The ocean is also a large source of halogens, which have far-reaching impacts on ozone, OH, NOx and VOC levels. Their emissions and chemistry should be implemented, as was done for example by Badia et al. (2019). Efforts should also be made to improve the representation of boundary conditions of the regional atmospheric models. FTIR column data were used in this work to adjust the boundary conditions for several compounds; other species should be adjusted, based on current knowledge relying on campaign data and network in situ measurements.

**Code availability.**

The WPS and WRF-Chem model code is provided by the National Center for Atmospheric Research (NCAR) (https://doi.org/10.5065/D6MK6B4K, NCAR, 2020). The WRF-Chem preprocessing tools can be found at https://www2.acom.ucar.edu/wrf-chem/wrf-chem-tools-community (last access: 1 October 2024, NCAR, 2023). Python scripts used in this work can be provided upon request.



**Data availability.**

Geographical static data used in the preprocessing step of the WRF-Chem simulations was downloaded from the WRF Users Page, hosted by the University Corporation for Atmospheric Research (UCAR), which can be found at https://www2.mmm.ucar.edu/wrf/users/download/get_sources_wps_geog.html (last access: 22 October 2024, UCAR, 2020). CAM-Chem files are distributed by NCAR and available at https://www.acom.ucar.edu/cam-chem/cam-chem.shtml (last access: 22 October 2024; https://doi.org/10.5065/NMP7-EP60; Buchholz et al., 2019). CAMS global reanalyses, provided by

the Copernicus Atmosphere Monitoring Service, were taken from https://ads.atmosphere.copernicus.eu/cdsapp#!/dataset/cams-global-reanalysis-eac4?tab=form (last access: 22 October 2024; Inness et al., 2019). Global emissions from the Emission Database for Global Atmospheric Research (EDGAR) v6.1, published by the Joint Research Center (JRC) at the European Commission, are available at https://data.jrc.ec.europa.eu/dataset/df521e05-6a3b-461c-965a-b703fb62313e (last access: 22 October 2024; Monforti

Ferrario et al., 2022). Speciated NMVOC emissions were obtained at https://data.jrc.ec.europa.eu/dataset/jrc-edgar-edgar_v432_voc_spec_timeseries (last access: 22 October 2024; Janssens-Maenhout et al., 2017). Temporal profiles for the EDGAR inventories have been detailed by Crippa et al. (2020), provided by the JRC and accessed at https://edgar.jrc.ec.europa.eu/dataset_temp_profile (last access: 22 October 2024; Crippa et al., 2020). The population density map for the region of Réunion Island is publicly available at

https://public.opendatasoft.com/explore/dataset/population-francaise-par-departement-2018/table/?disjunctive.departement (last access: 22 October 2024; French National Institute of Statistics and Economic Studies (INSEE), 2018). Air quality measurements of surface $NO_2$, NO and $O_3$ were obtained from the Atmo-Réunion website at https://atmo-reunion.net/ (last access: 22 October 2024). The PTR-MS measurements are available at https://data.aeronomie.be/dataset/long-term-in-situ-o-voc-measurements-at-the-maido-observatory-reunion-island-v2 (last access: 22 October 2024; Amelynck et al., 2024). FTIR

observations at the Maïdo Observatory can be accessed at https://www-air.larc.nasa.gov/missions/ndacc/data.html?station=la.reunion.maido/hdf/ftir/ (last access: 22 October 2024), except for PAN (which can be provided on request). TROPOMI S5P V3.2 data, as well as TM5 profiles, can be accessed at the ESA's public dataspace https://browser.dataspace.copernicus.eu/ (last access: 22 October 2024; European Space Agency, 2023).

**Author contributions.**

CP was responsible for running the WRF-Chem simulations, compiling the necessary data, conducting the model validation, and writing the initial version of the manuscript. JFM and TS developed the project concept, oversaw the work, and contributed to interpreting the findings on a regular basis. CA, BWDV and NS provided the PTR-MS measurements, refined the relevant methodology text and helped with clarifications regarding comparisons with this data. CV and NK provided the FTIR data. CV wrote the text describing the FTIR methodology. JB supplied the 100m resolution dataset of plant functional

types and isoprene emission factors. CMV and PT produced the 1 km resolution anthropogenic emission inventory. JFM





handled the revision and editing of the manuscript. CP implemented co-author comments and edits and formatted the final version of the manuscript.

**Competing interests.**

The authors declare that they have no conflict of interest.

**Acknowledgements.**

We thank Guillaume Peris and Dr. Marion Haramboure from Atmo-Réunion for helping us access ancillary air quality data from the Atmo-Réunion website.

**Financial support.**

This research received funding from the Belgian Federal Science Policy Office (Belspo) through the European-Space-
Agency-funded ProDex TROVA-E2 (2020-2023) and TROVA-3 (2024-2025) projects, as well as via the OCTAVE project (grant no. BR/175/A2/OCTAVE) also funded by Belspo. This work was also supported by the EU Horizon 2020 programme (grant no. ACTRIS-2, 654109).

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
