# Peer review of "Constraining the budget of NOx and VOCs at a remote Tropical island using multi-platform observations and WRF-Chem model simulations"

_EGUsphere, 2024_

## Author Comment (AC1)

**Reviewer Comment #1**

This is a quite complete paper about the atmospheric chemistry over the Reunion Island. The focus is placed on $NO_x$ ($NO+NO_2$) and VOC (volatile organic compounds). The period used extended during January and July 2019. Meteorological variables together with atmospheric concentrations were used. Moreover, the comparison between measured and calculated values is presented. A detailed description of input data is presented and results cover formaldehyde, methanol, isoprene, monoterpenes, acetaldehyde, acetone among other substances. Calculated series are presented, but also the daily cycle is highlighted. Finally, FTIR and TROPOMI data are compared with modelled results. Consequently, the paper presents a noticeable amount of information of atmospheric chemistry for this remote site, and it merits to be published in Atmospheric Chemistry and Physics, although some minor changes should be introduced prior to its final acceptance.

Although the analysis depth is noticeable, the time extension is quite limited. Perhaps the authors could introduce some comments about the data representativeness, i.e., they could indicate if atmospheric conditions are steady over this site or if accused and frequent changes have been observed.

- *In this study, the simulation periods (January and July 2019) represent distinct seasons, austral summer and winter, respectively. These months capture the seasonal extremes for this region, with other months generally falling within the range of these meteorological conditions. However, the same does not apply to extreme weather events, such as cyclones, which bring intense precipitation and strong winds, significantly altering atmospheric conditions and model results. Clearly, our model simulations cannot represent such extreme situations.*

Similarly, since this is a quite limited site, the authors could inform about the possible extension of their results to other sites around the world. This information would be useful to increase the result applicability and the number of possible readers.

- *In many respects, Maïdo is unique, due to the mountainous terrain and relative isolation of the island and the particular distribution of anthropogenic, biogenic and oceanic sources around the site. Therefore, to a large degree, the comparisons inform us mostly on the model performance for this specific site. Nevertheless, as discussed in the manuscript, several results provide information relevant to wider areas, e.g. the large underestimation of key OVOCs (e.g. acetic acid and acetaldehyde), the good performance for isoprene and its oxidation products, and the confirmation of a significant biogenic source of MEK. Quantifying the model biases for (O)VOCs is of great importance due to their influence for OH reactivity on a local scale, and this understanding is essential for improving models globally (e.g. Read et al., 2012).*

  *Read, K. A., Carpenter, L. J., Arnold, S. R., Beale, R., Nightingale, P. D., Hopkins, J. R., Lewis, A. C., Lee, J. D., Mendes, L. and Pickering, S. J.: Multiannual observations of acetone, methanol, and acetaldehyde in remote tropical Atlantic Air: Implications for atmospheric OVOC budgets and oxidative capacity, Environmental Science & Technology, 46(20), 11028–11039, doi:10.1021/es302082p, 2012a.*

Most of the paper is focused on Reunion Island. However, section 3.5 includes Mauritius Island. Perhaps the authors could introduce a short comment for this change in the studied area. Moreover, they should indicate if input data belong to Reunion Island and if model results are extended to Mauritius Island.

- *As explained in the manuscript, the model domain encompasses Réunion, Mauritius, part of Madagascar and a large portion of the Indian Ocean. The lateral boundary condition data relate to this large domain, and therefore not specifically to Reunion Island. As explained in the paper, the emission inventory is a mix of high-resolution data over Réunion and low-resolution data elsewhere (including Mauritius). All model results are produced on the same larger domain (denoted d01), even though most results are shown for Réunion Island only, since it is the focus of our study. When considering the $NO_2$ columns over the entire modelled domain, we noted a significant outflow from Mauritius, which we thought was worth showing in the paper. We therefore extended the comparison over Mauritius to check its influence on Réunion Island NO2 and also evaluate the quality of the global inventory. The comparison clearly shows an overestimation of Mauritius emissions, as discussed in the text.*

Possible weaknesses or limitations of this study could be introduced.

- *This is covered in the conclusion (Lines 1129-1141).*

Minor remarks.

Perhaps citations of figures and previous studies should not be introduced in the conclusion section.

- *This has been amended.*

**Reviewer Comment #2**

**General Comments**

The manuscript evaluates the WRF-Chem model's ability to simulate chemical species over Réunion Island, focusing on key species such as formaldehyde, methanol, isoprene, Iox (isoprene oxidation products), monoterpenes, acetone, $NO_2$, $NO_x$, OH, and $O_3$. While the study offers valuable insights into VOC and $NO_x$ simulations in a tropical setting, significant improvements are needed, particularly in defining clear scientific objectives and providing a robust statistical evaluation of model performance for both meteorological and chemical species. Substantial revisions are required.

**Major Comments:**

1. The manuscript provides excessive detail on data collection methods, such as PTR-MS measurements. Given the primary focus on model evaluation, these descriptions should be condensed, retaining only essential information relevant to data analysis and model validation.

- *We agree that the PTR-MS measurement description could be shortened. Our main focus being on model evaluation, however, we think important to keep sufficient information on the method, since many details are highly relevant to the model validation (e.g. interferences). We shortened the PTR-MS methodology section, and referred to the measurement report (Verreyken et al., 2021).*

2. The study introduces updates to chemical mechanisms (Sections 2.2.3.1, 2.2.3.2, Table 2, and sensitivity run S4). If these updates represent novel developments, they should be explicitly highlighted in the abstract and conclusion. Otherwise, proper citations should be included.

- *These updates are novel developments. They are mentioned in the abstract (line 26), but not in the conclusions. We have therefore added the following sentence in this section: "In addition, the chemical mechanism of the model was updated to better account for OH recycling in isoprene oxidation and for MEK degradation mechanisms."*

3. Despite significant differences in simulated precursor concentrations (Fig. 18), $O_3$ predictions remain relatively stable. The authors should provide a thorough explanation of this discrepancy.

- *The ozone surface concentrations actually do show a significant variation between the two seasons (almost a factor of 2, see Fig. 9). The stability of $O_3$ predictions is therefore quite relative. The strong variations in precursor concentrations (specifically NOx), can be attributed to the longer lifetime of NOx in July (winter) compared to January (summer), as discussed in Section 3.5. Ozone might be relatively stable because both the production and loss of ozone are reduced during winter, partly due to lower photolysis rates and H2O levels (which promote ozone loss and radical production). The ozone budget is complex (Pandis-Seinfeld book or another reference). Since it is not the focus of this work, we prefer to keep the discussion short.*

4. The use of sequential 2-day simulations (Lines 158–160) is unsuitable for evaluating monthly trends. It is recommended to conduct daily 48-hour simulations, starting at a consistent time (e.g., 00Z or 12Z UTC), and assess either Day 1 or Day 2 results to account for numerical model performance variations over different forecast hours.

- *Our choice of sequential 2-day simulations was motivated mostly by computational constraints. While running independent daily 48-hour simulations with a fixed initialization time can minimize variations in the model performance, it would also significantly increase the computational cost without necessarily improving the representation of monthly trends. In previous work, we conducted similar subsequent simulations, but considering a spin-up time (Poraicu et al., 2023). When we tested this for Reunion Island, we saw no discontinuity in the results, and therefore sacrificed the 6-hour overlap to minimize the computational burden.*
- *Nevertheless, the reviewer comment does point to a potential error. To test this, we reran the R0 simulation for the month of January 2019, starting the simulation on the $2^{nd}$ of January. In this test, day 1 and day 2 of each 48-hour run correspond to even and odd days of the month, respectively, whereas the reverse sequence was used in the simulations shown in the paper.*
- *We mostly looked at the average diurnal profile of the compounds measured by the PTR-MS, which is the key comparison used in our work to adjust the emissions and evaluate the model. The results show a performance that is consistent with our current presented results, with a few notable differences. For example, isoprene shows a slightly different profile (~10% difference near noon, and ~20% difference near midnight, when values are near- zero).*

[Figure]

- *We therefore conclude that the simulations conducted in this work are sufficiently robust to comment on the necessary adjustments of WRF-Chem for the scope of this study. We encourage future studies using WRF-Chem to conduct a similar analysis of the numerical performance, assessing the trade-off between computational cost and accuracy.*

5. The description of biogenic emissions in Section 2.3.2 should be streamlined unless significant modifications were made to MEGAN 2.0.4.

- *Since we adjusted some of the factors used in the MEGAN 2.0.4 algorithm code in WRF-Chem, we felt that it was necessary to explicitly define how the emissions are calculated, and where these factors come into use. Nevertheless, we have streamlined the description of the response factors to environmental conditions and simply reference the MEGAN algorithm paper (Guenther et al., 2006).*

  *Guenther, A., Karl, T., Harley, P., Wiedinmyer, C., Palmer, P. I., and Geron, C.: Estimates of global terrestrial isoprene emissions using MEGAN (Model of Emissions of Gases and Aerosols from Nature), Atmos. Chem. Phys., 6, 3181–3210, https://doi.org/10.5194/acp-6-3181-2006, 2006.*

6. Sensitivity run S1: The over-prediction of $NO_2$ and $NO_x$ at the LP station (Lines 598–600) is attributed to surface-level emission injection. Instead of reducing $NO_x$ emissions by a factor of five, a more appropriate approach would be to inject emissions at the plume rise height.

- *Indeed, we did not inject the emissions at the plume rise height, mostly due to the lack of vertical distribution information in the emission inventory we used for this region. While emission inventories may provide estimates (e.g. as was the case in Poraicu et al., 2023), true injection heights can be much higher due to plume rise effects (e.g. Bieser et al., 2011). Furthermore, our model comparisons with TROPOMI $NO_2$ columns, which are more sensitive to pollutants present at higher altitudes, indicated a significant overestimation of the modelled columns when adopting the unscaled inventory emissions (Fig. S7). Placing emissions higher would therefore worsen the agreement. This suggests that the factor of 5 reduction of emissions is justified.*

  *Bieser, J., Aulinger, A., Matthias, V., Quante, M. and Denier van der Gon, H. A. C.: Vertical Emission Profiles for Europe based on plume rise calculations, Environmental Pollution, 159(10), 2935–2946, doi:10.1016/j.envpol.2011.04.030, 2011.*

  *Poraicu, C., Müller, J.-F., Stavrakou, T., Fonteyn, D., Tack, F., Deutsch, F., Laffineur, Q., Van Malderen, R., and Veldeman, N.: Cross-evaluating WRF-Chem v4.1.2, TROPOMI, APEX, and in situ $NO_2$ measurements over Antwerp, Belgium, Geosci. Model Dev., 16, 479–508, https://doi.org/10.5194/gmd-16-479-2023, 2023.*

7. Figure 8: Evaluating meteorological simulations based on a single-site comparison is insufficient. Are additional meteorological observation sites available? Including statistical metrics (e.g., correlation coefficient, RMSE, mean bias) would strengthen the evaluation.

- *Our manuscript already includes a statistical analysis of meteorological comparisons at the Maïdo station, including the aforementioned metrics (see Table S2). Our comparison with meteorological measurements at Maïdo was essential to the scope of our study, which compares modeled results with PTR-MS measurements at the same location. This was particularly relevant for evaluating the simulation results for biogenically emitted and short-lived compounds, which are strongly dependent on local meteorological conditions, like*

*temperature, radiation and winds. We agree that a full evaluation of modelled meteorology would require a broader set of observations and metrics over a wider area, but, especially since the manuscript is already very long, we consider that the Maïdo comparisons are sufficient for our purposes.*

8. Figure 10: Include a simulated vs. observed $O_3$ comparison at the Le Port station to assess the impact of $NO_x$ and VOC predictions on surface $O_3$ levels.

- *There are no observations of $O_3$ at the Le Port station, or nearby. All the existing $O_3$ measurements over Reunion Island are categorized in the "other stations" and included in this work already.*

9. Lines 368–369 mention 18 air quality monitoring stations. The evaluation should incorporate all available sites with statistical metrics such as correlation coefficient, root mean square error (RMSE), and index of agreement (IOA).

- *Indeed, there are 18 air quality monitoring stations located in Reunion Island, however only some of them measure compounds that we are interested in (NOx and O3) during our study period. The 12 remaining stations included in this work, where NOx and/or $O_3$ are measured and available, are shown in Figure S2. We now provide in the Supplement a statistical analysis for the R0 run, for both months, for the available species at every station.*

10. Lines 33–34: The statement attributing a 6 ppbv $O_3$ overestimation to the exclusion of halogen chemistry is overly simplistic. Ozone overprediction can stem from multiple factors, including uncertainties in emissions, vertical mixing, model resolution, and atmospheric chemistry.

- *We agree that multiple factors can contribute to ozone overestimation. Still, we have identified a specific missing process in the model (halogen chemistry) which might quantitatively account for the discrepancy, and this should be obviously pointed out (e.g. Sander and Crutzen, 1996; Read et al., 2008; Saiz-Lopez et al., 2012). We know that these processes are missing from the standard WRF-Chem model, but studies like Badia et al. (2019) have implemented them, finding ozone removal by halogens can reach 25–60%. We have amended the text in this section to stress the possible importance of other model factors, possibly contributing to the model bias.*

*Badia, A., Reeves, C. E., Baker, A. R., Saiz-Lopez, A., Volkamer, R., Koenig, T. K., Apel, E. C., Hornbrook, R. S., Carpenter, L. J., Andrews, S. J., Sherwen, T., and von Glasow, R.: Importance of reactive halogens in the tropical marine atmosphere: a regional modelling study using WRF-Chem, Atmos. Chem. Phys., 19, 3161–3189, https://doi.org/10.5194/acp-19-3161-2019, 2019.*

*Read, K. A., Mahajan, A. S., Carpenter, L. J., Evans, M. J., Faria, B. V., Heard, D. E., Hopkins, J. R., Lee, J. D., Moller, S. J., Lewis, A. C., Mendes, L., McQuaid, J. B., Oetjen, H., Saiz-Lopez, A., Pilling, M. J. and Plane, J. M.: Extensive halogen-mediated ozone destruction over the tropical Atlantic Ocean, Nature, 453(7199), 1232–1235, doi:10.1038/nature07035, 2008.*

*Saiz-Lopez, A., Lamarque, J.-F., Kinnison, D. E., Tilmes, S., Ordóñez, C., Orlando, J. J., Conley, A. J., Plane, J. M. C., Mahajan, A. S., Sousa Santos, G., Atlas, E. L., Blake, D. R., Sander, S. P., Schauffler, S., Thompson, A. M., and Brasseur, G.: Estimating the climate significance of halogen-driven ozone loss in the tropical marine troposphere, Atmos. Chem. Phys., 12, 3939–3949, https://doi.org/10.5194/acp-12-3939-2012, 2012.*

*Sander, R. and Crutzen, P. J.: Model study indicating halogen activation and ozone destruction in polluted air masses transported to the sea, Journal of Geophysical Research: Atmospheres, 101(D4), 9121–9138, doi:10.1029/95jd03793, 1996.*

**Minor Comments:**

1. Line 23: Define "a.s.l."

- *"a.s.l." = above sea level, amended.*

2. Line 12: Add Piton de la Fournaise's location to Figure 3.

- *Amended.*

3. Line 176: Define MEGAN and include references.

- *Done.*

4. Line 190: Spell out MVK and MACR.

- *Amended.*

5. Line 230: Clarify the meaning of "higher resolution data (0.75° x 0.75°)."

- *It only means that CAMS (with 0.75˚ x 0.75˚ resolution) has higher resolution than CAM-CHEM (0.9 x 1.25°).*

6. Line 233: Spell out "BIGALK."

- *Amended.*

7. Line 241: Define "EDGAR."

- *Amended.*

8. Line 242: Define "HTAP."

- *Amended.*

9. Line 354: Avoid duplicate definitions (e.g., ISOPOOH on Line 196).

- *Done.*

10. Line 560: Define $NO_x$ (=$NO+NO_2$) upon first mention.

- *This was defined in the abstract.*

11. Lines 578–579: List stations categorized as "other stations."

- *As stated in the legend of Fig. 10, this refers to stations 3-9 of Table S3.*

12. Line 582: Correct "cyan" to "blue" (also check Line 588).

- *Amended.*

13. Line 591: No green dots appear in Figure 2—did you mean red dots?

- *Yes, this is corrected in an amended version of the document.*

14. Lines 613–614: The statement that $NO_x$ model agreement issues stem from $O_3$ overestimation is confused, as $NO_x$ is one of major precursors of $O_3$ formation rather than being driven by $O_3$ levels.

- *Of course, NOx is a precursor of ozone and is important in dictating ozone levels. Nevertheless, if $O_3$ levels are overestimated, then the conversion of NO to $NO_2$ through the NO + $O_3$ is too fast, leading to an overestimated $NO_2$ to NO ratio. Therefore, while NOx emissions control $O_3$ formation, an overestimation of $O_3$ can, in turn, introduce a bias to the modeled $NO_2$/NO ratio.*

15. Line 629: Should reference be "Fig. 14"?

- *Yes, Figure 14 shows the diurnal profiles of modelled and observed compounds at the Maïdo Observatory, as the text indicates.*